# Training-Free Loosely Speculative Decoding: Accepting Semantically Correct Drafts Beyond Exact Match

**Jinze Li**[1,2], **Yixing Xu**[1], **Guanchen Li**[1], **Shuo Yang**[2], **Jinfeng Xu**[2], **Xuanwu Yin**[1],
**Dong Li**[1], **Edith C.H. Ngai**[2,*] **Emad Barsoum**[1]
[1]Advanced Micro Devices, Inc., Beijing, China
[2]Department of Electrical and Electronic Engineering, The University of Hong Kong
`{lijinze-hku, shuo.yang, jinfeng}@connect.hku.hk`
`{yixing.xu, Xuanwu.Yin, d.li, emad.barsoum}@amd.com`
`chngai@eee.hku.hk`

## Abstract

Large language models (LLMs) achieve strong performance across diverse tasks but suffer from high inference latency due to their autoregressive generation. Speculative Decoding (SPD) mitigates this issue by verifying candidate tokens in parallel from a smaller draft model, yet its strict *exact-match verification* discards many semantically valid continuations. Moreover, existing training-based SPD methods often suffer from performance degradation on out-of-distribution (OOD) tasks. To this end, we propose **Training-Free Loosely Speculative Decoding (FLy)**, a novel method that loosens the rigid verification criterion by leveraging the target model's self-corrective behavior to judge whether a draft–target mismatch remains semantically valid. FLy introduces a two-tier mechanism: an **entropy-level gate** that identifies whether the current token allows multiple plausible alternatives or is nearly deterministic, and a **token-level deferred window** that distinguishes genuine errors from *differently worded yet semantically correct* variants. To further reduce latency, we design a **multi-level acceleration** strategy that accelerates not only the target model but also the drafter itself. Owing to its training-free design, FLy composes seamlessly with arbitrary draft–target pairs and generalizes across models and domains without hyperparameter re-tuning. Experiments show that FLy preserves ≥**99%** of the target model's accuracy while achieving an average **2.81×** speedup on Llama-3.1-70B-Instruct and **5.07×** speedup on the 405B variant. Notably, on out-of-domain datasets, our method remains highly effective and outperforms the training-based method EAGLE-3 by **1.62×**. Our code is available at https://github.com/AMD-AGI/FLy.

## 1 Introduction

While Large Language Models (LLMs) (Vaswani et al., 2017) have demonstrated impressive capabilities (Dubey et al., 2024; Anil et al., 2023; DeepSeek-AI et al., 2025), their auto-regressive inference strategy entails substantial latency, which becomes more pronounced as model size scales. Speculative decoding (SPD) (Stern et al., 2018) has emerged as a promising solution to accelerate generation without compromising quality. In SPD, a lightweight draft model proposes multiple candidate tokens sequentially, which the larger target model verifies in parallel, accepting those consistent with its own predictions. This procedure provably preserves the target model's distribution while delivering significant throughput gains.

However, standard SPD is fundamentally constrained by its exact-match rule: the target accepts a draft token only if it is identical to its own generation. This rigid requirement forces the rejection of many plausible continuations, even those semantically aligned, thereby discarding useful tokens and limiting speedup. Recent work (Bachmann et al., 2025) shows that even high-quality drafts

---

*Corresponding to `chngai@eee.hku.hk`.

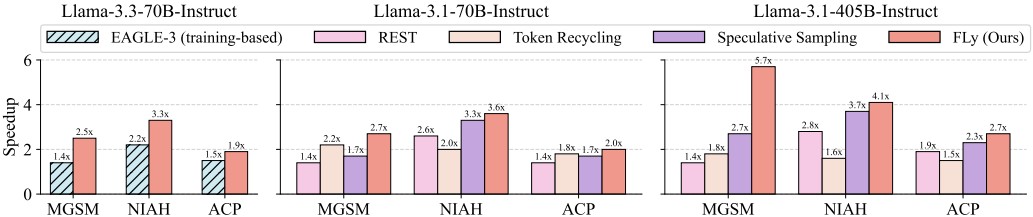

Figure 1: Speedup on out-of-domain (OOD) datasets. Training-based method EAGLE-3 suffers significant degradation under OOD conditions. Our approach surpasses existing methods across datasets and models by accepting more semantically valid tokens, achieving SOTA performance.

(*e.g.*, human-written text) achieve low acceptance under this scheme. As draft models continue to improve, such strict rejection becomes an increasingly inefficient bottleneck, highlighting the importance of more flexible verification strategies. Existing works have proposed "loose" variants of SPD that relax the strict verification rule. A representative example is JudgeDecoding (Bachmann et al., 2025), which trains an auxiliary classifier to decide whether a draft token is contextually valid. Although effective, this approach requires carefully curated training data and training, incurring high annotation costs. Moreover, the supervisively trained classifier often fails to generalize across multiple domains or tasks, making the method brittle in out-of-distribution (OOD) settings.

To address these limitations, we propose **Training-Free Loosely Speculative Decoding (FLy)**, which relaxes verification by accepting semantically correct drafts without additional training. The central insight is that LLMs tend to exhibit self-corrective behavior when conditioned on genuinely erroneous tokens, but not when faced with merely *worded differently yet semantically valid* alternatives (Pan et al., 2023; Bachmann et al., 2025). Building on this property, FLy leverages the target model's own behavior to distinguish harmful mismatches from semantically equivalent continuations. Specifically, FLy introduces a two-tier mechanism for mismatch handling. The first component is an **entropy-level gate**, which classifies the mismatch position as either ambiguous (multiple valid alternatives) or deterministic (a single plausible token) based on the target model's token-level entropy. If the entropy falls below a threshold, indicating a near-deterministic case such as a numerical calculation, the mismatch is immediately rejected. Conversely, if the entropy exceeds the threshold, FLy activates a **token-level deferred window** spanning the next several tokens. Within this window, the mismatch is provisionally accepted. If another mismatch emerges, it signals the model's attempt to *course-correct* an earlier error, and the initial token is retroactively rejected. Otherwise, the token is deemed a semantically valid continuation and retained.

By accepting semantically correct mismatches, the average number of accepted tokens ($\tau$) rises markedly. Thus, the drafter needs to propose a larger set of tokens per round, which raises the drafter's generation cost to the point where it becomes non-negligible compared to common SPD methods (Cai et al., 2024; Li et al., 2025b), thereby diminishing overall speedups. To mitigate this problem, we propose a **multi-level acceleration** scheme that not only accelerates the target model, but also speeds up the drafter itself, preventing the drafting stage from becoming the dominant bottleneck as $\tau$ grows.

Notably, FLy is a plug-and-play method requiring no data collection or training, and it is model-agnostic. A single drafter can accelerate different targets, and distinct drafters can be paired with the same target without retraining. Moreover, FLy is inherently robust to distribution shifts, since it does not depend on any training data whose generalization may fail on OOD test data (*cf.* Figure 1). Experimental results demonstrate that FLy achieves an average **2.53×** speedup on out-of-domain datasets (**1.62×** faster than EAGLE-3) and **2.69×** on in-domain datasets (**1.05×** faster than EAGLE-2) with 70B-scale models, and an average of **5.07×** speedup with 405B-scale models. In all circumstances, FLy preserves ≥**99%** of target accuracy on 70B and 405B models.

## 2 METHOD

In this section, we first introduce the preliminaries of Speculative Decoding (SPD) in Section 2.1. Then, we propose **Training-Free Loosely Speculative Decoding (FLy)** in Section 2.2, a method that relaxes the rigid *exact match* verification rule used in standard SPD without requiring any training, as shown in Figure 2. Different from existing mechanisms, FLy can accept *worded-differently but semantically correct* tokens, thereby improving the mean accepted tokens ($\tau$) and the speedup

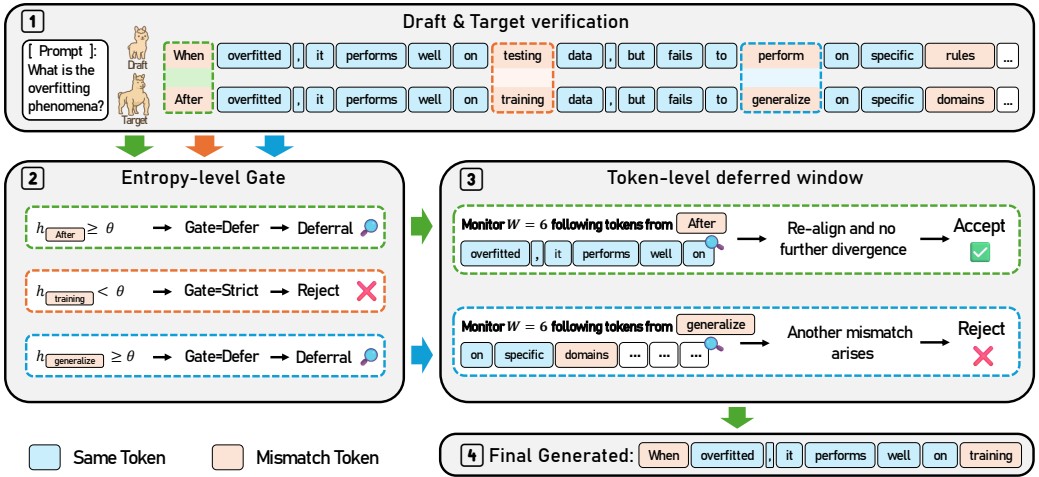

Figure 2: Overview of our proposed FLy. (1) When the draft and target tokens differ, we do not immediately reject the case as in prior SPD methods. Instead, our two-tier scheme assesses whether the mismatch is semantically valid and rejects only truly invalid cases. (2) An entropy gate rejects deterministic target predictions where $h < \theta$, deferring ambiguous mismatches. (3)A token-level deferral window ($W = 6$) then monitors for continued divergence. (4) The final generation demonstrates that FLy admits more semantically valid continuations, whereas standard SPD would reject at the first mismatch.

ratio with minimum accuracy drop. The detailed algorithm is provided in Appendix A. Finally, a multi-level acceleration method is used to further boost the final speed-up ratio in Section 2.3.

## 2.1 PRELIMINARIES

In each SPD round $t$, a drafter $\mathcal{M}_D$ sequentially proposes $K$ draft tokens $\{\hat{y}_i^t\}_{i=1}^K$, and the target model $\mathcal{M}_T$ verifies them together with the last generated token in round $t-1$ (denoted as $y_{-1}^{t-1}$) in parallel by performing a single forward pass on the draft tokens to produce logits $\{\ell_i^t\}_{i=1}^{K+1} = \mathcal{M}_T([y_{-1}^{t-1}, \{\hat{y}_i^t\}_{i=1}^K]) \in \mathbb{R}^{|\mathcal{V}|}$, where $\mathcal{V}$ denotes the set of the vocabulary and $|\mathcal{V}|$ the vocabulary size. The token in position $K+1$ is the bonus token. The target model then normalizes the logits with a softmax function, and selects the token with the highest normalized logit as the output token:

$$p_{\mathcal{M}_T,i}(v) = \text{softmax}(\ell_i^t)_v, \quad y_i^t = \arg\max_{v \in \mathcal{V}} p_{\mathcal{M}_T,i}(v), \tag{1}$$

where $p_{\mathcal{M}_T,i}(\cdot)$ denotes the normalized predictive distribution from the target model $\mathcal{M}_T$ over the vocabulary set $\mathcal{V}$ at position $i \in [1, K+1]$, and $y_i^t$ denotes the corresponding output token of $\mathcal{M}_T$.

Then, we define the match indicator $\Delta_i^t$ to judge whether the prediction at position $i$ of the draft model matches that of the target model:

$$\Delta_i^t = \mathbb{1}[\hat{y}_i^t = y_i^t], \ 1 \leq i \leq K \tag{2}$$

where $\mathbb{1}[\cdot]$ denotes the *indicator function*, which equals 1 when its condition is true and 0 otherwise. Concretely, $\hat{y}_i^t = y_i^t$ means that the output tokens from draft and target models coincide at position $i$, leading to $\Delta_i^t = 1$, and vice versa. We omit the superscript $t$ in the following paragraph and only use $\Delta_i$ for simplicity.

The verification strategy of standard SPD accepts all draft tokens until the first position where a mismatch occurs, and discards the rest of the generated draft tokens. Then, it will accept one more token generated from the target model as a bonus. Formally, in each draft round, the number of accepted tokens $s_{\text{standard}}$ can be defined as:

$$s_{\text{standard}} = \begin{cases} \min\{i \mid \Delta_i = 0\}, & \prod_{i=1}^K \Delta_i = 0, \\ K+1, & \text{otherwise.} \end{cases} \tag{3}$$

## 2.2 TRAINING-FREE LOOSELY SPECULATIVE DECODING (FLY)

FLy modifies only the verification strategy of standard SPD. Given $K$ draft tokens in a draft round, there will be several mismatches that occur at different positions:

$$J = \{j \mid \Delta_j = 0\}, \ \ 1 \le j \le K. \tag{4}$$

For each mismatch position $j \in J$, we compute a normalized entropy given the normalized logits from the target model at that position:

$$h_j \ = \ \frac{-\sum_{v \in \mathcal{V}} p_{\mathcal{M}_T,j}(v) \log p_{\mathcal{M}_T,j}(v)}{\log |\mathcal{V}|}. \tag{5}$$

The entropy $h_j \in [0,1]$ quantifies the target model's uncertainty at position $j$. A large $h_j$ indicates that several tokens could serve as valid alternatives at this position, whereas a small $h_j$ indicates that the target model tends to accept only the top-rated token.

**Entropy-level gate.** Given Equation. 5, we apply an entropy gate at the mismatch position $j$ to decide whether the current mismatch should be directly rejected. Specifically, we define an entropy threshold $\theta \in [0,1]$, and the gate serves as a lightweight per-token ambiguity detector driven by the target model. If the target model is confident (*i.e.,* $h_j < \theta$), indicating a strong preference for its top-rated token such as digits in arithmetic, we fall back to the standard SPD rule and reject all tokens starting from the mismatch position $\{\hat{y}_i^t\}_{i=j}^K$. Otherwise, the token at this position is ambiguous (*i.e.,* $h_j \ge \theta$), meaning that several tokens are nearly interchangeable such as coreferent pronouns, and we activate a token-level deferred window mechanism which will be introduced later.

Concretely, at the mismatch position $j$, we apply a Gate with a threshold $\theta$ to decide whether deferral should be invoked:

$$\mathsf{Gate}(j) = \begin{cases} \mathsf{Strict}, & h_j < \theta, \\ \mathsf{Defer}, & h_j \ge \theta, \end{cases} \tag{6}$$

where Strict refers to the standard SPD rule that rejects tokens after position $j$, and Defer means using the token-level deferred window, postponing the rejection decision and observing the target's behavior over the following few tokens. This separation prevents deferral at positions where an incorrect token would likely corrupt exactness. When treating the Defer operation as a temporal acceptance, the number of accepted tokens with entropy-level gate $s_{\mathrm{gate}}$ can be formulated as:

$$s_{\mathrm{gate}} = \begin{cases} \min_{j \in J}\{ j \mid \mathsf{Gate}(j) = \mathsf{Strict} \}, & \exists j \in J : \mathsf{Gate}(j) = \mathsf{Strict}, \\ K+1, & \text{otherwise.} \end{cases} \tag{7}$$

**Token-level deferred window.** When Defer is active at the mismatch index $j$, FLy uses a look-ahead window spanning $W$ tokens to decide whether the mismatch is benign.

Over the next $W$ tokens, we compute the total number of mismatches:

$$N_W(j) \ = \ \sum_{i=j+1}^{j+W} (1 - \Delta_i). \tag{8}$$

Then, we can determine to accept or reject this mismatch at position $j$ based on whether another mismatch appears in this window:

$$\mathsf{DeferDecide}(j) = \begin{cases} \mathsf{Accept}, & h_j \ge \theta, \ N_W(j) = 0 \text{ and } j + W \le K \\ \mathsf{Reject}, & \text{otherwise.} \end{cases} \tag{9}$$

Our decision rule leverages the target LLM's own behavior. When conditioned on a semantically invalid token, it typically exhibits corrective behavior in its subsequent generations. Specifically, if the target model can continue generating from the mismatched token without further divergence, the first divergent token is likely semantically correct and can be accepted. Otherwise, if the target keeps disagreeing, attempting to *course-correct* the mismatch and thereby causing another divergence, we reject the mismatch at $j$ and the following draft tokens.

Note that we only have a maximum of $K$ draft tokens in each round. Thus, $j + W > K$ indicates that we get into the boundary case where the first mismatch index $j$ falls within the last $W$ positions

and the size of the look-ahead window would be less than $W$. Then, we immediately reject this mismatch at position $j$ since the rest of the tokens are not enough to judge the semantic correctness of this mismatch. It is worth noting that this immediate rejection incurs negligible cost, since only a few trailing tokens are discarded. We adopt this conservative rule to preserve accuracy and keep the implementation simple.

The number of accepted tokens with deferred window $s_{\text{defer}}$ can be formulated as:

$$s_{\text{defer}} = \begin{cases} \min_{j \in J}\{ j \mid \mathsf{DeferDecide}(j) = \mathsf{Reject}\}, & \exists j \in J : \mathsf{DeferDecide}(j) = \mathsf{Reject}, \\ K + 1, & \text{otherwise.} \end{cases} \tag{10}$$

In each SPD round, the final output of FLy depends on where the first rejection occurs. Therefore, the number of accepted tokens with our proposed FLy $s_{\text{FLy}}$ can be formulated as:

$$s_{\text{FLy}} = \min\{s_{\text{gate}}, s_{\text{defer}}\}. \tag{11}$$

As we index SPD rounds by $t \in \{1, 2, \ldots, T\}$, we define $s_{\text{FLy}}^t$ as the number of tokens accepted at round $t$. The mean number of accepted tokens $\tau$ over the run is then given by:

$$\tau = \frac{1}{T} \sum_{t=1}^{T} s_{\text{FLy}}^t. \tag{12}$$

Notably, FLy introduces *no additional forward passes*. Instead, it computes per-token entropy directly from the already-available logits. As a result, the computation overhead is negligible compared to the inference cost of the draft and target model, as shown in the last line in Table 1.

## 2.3 MULTI-LEVEL ACCELERATION

By leveraging the entropy-level gate and token-level deferred window, FLy attains substantially longer acceptance length ($\tau$). This lengthening reduces the frequency of target validations and hence their proportion of total latency, which in turn amplifies the draft model's auto-regressive cost as the dominant runtime bottleneck (Table 1). This observation motivates our further contribution: a **Multi-Level Acceleration (MLA)**

| Components | L3 70B | | L3 405B | |
|---|---|---|---|---|
| | w/o MLA | w/ MLA | w/o MLA | w/ MLA |
| Draft time | 245.51 | 197.45 | 428.30 | 363.01 |
| Target verification | 58.62 | | 200.97 | |
| Gate & Window activation | 0.45 | | 0.57 | |

Table 1: Wall time (ms) of different components for each SPD round.

strategy that applies speculative acceleration not only to the target model (as in standard SPD) but also to the draft model. By reducing overhead at drafting stage, MLA achieves greater end-to-end efficiency.

A few prior works (Georganas et al., 2025; Sun et al., 2024) have investigated multi-level acceleration, but they generally depend on parameterized mini-draft models, such as quantized variants or small language models (SLMs). To ensure broader applicability, we design the draft-acceleration stage in MLA to be parameter-free and plug-and-play, allowing seamless integration with the proposed FLy. Specifically, we instantiate MLA with Prompt Lookup Decoding (PLD) (Saxena, 2023), a simple $n$-gram retrieval method that is extremely fast and training-free, thereby avoiding additional bias and preserving cross-domain generalization. Consequently, MLA reduces draft-side overhead and achieves higher end-to-end speedups, as illustrated in Table 1 and Table 5.

It is worth noting that MLA is orthogonal to the verification scheme and is primarily designed to speed up the drafting phase itself. It is particularly beneficial in our setting, where the algorithm can safely propose long draft sequences, making drafting time rather than target-model verification time the primary bottleneck. This stands in clear contrast to prior approaches, where the target model's verification phase often dominates the runtime.

## 3 EXPERIMENTS

We begin this section by outlining the experimental settings in Section 3.1. Then, we compare our proposed FLy against current SOTA methods in Section 3.2. Next, to underscore its capability

| Model | Training | Method | ACP prog_gen | | NIAH multivalue | | MGSM de | | fr | | th | | Mean | |
|---|---|---|---|---|---|---|---|---|---|---|---|---|---|---|
| | | | Speedup | $\tau$ | Speedup | $\tau$ | Speedup | $\tau$ | Speedup | $\tau$ | Speedup | $\tau$ | Speedup | $\tau$ |
| | | | | | | | Temperature=0 | | | | | | | |
| L33 70B | ✓ | EAGLE-3 | 1.52× | 3.92 | 2.15× | 2.43 | 1.57× | 2.23 | 1.72× | 2.65 | 0.84× | 1.26 | 1.56× | 2.50 |
| | ✗ | FLy (Ours) | **1.88×** | **10.95** | **3.34×** | **13.53** | **2.64×** | **11.41** | **2.18×** | **10.29** | **2.60×** | **10.58** | **2.53×** | **11.35** |
| L31 70B | ✗ | SpS | 1.69× | 10.94 | 3.33× | 15.82 | 1.79× | 9.23 | 1.62× | 8.72 | 1.78× | 9.84 | 2.04× | 10.91 |
| | ✗ | REST | 1.35× | 1.57 | 2.63× | 3.03 | 1.44× | 1.47 | 1.28× | 1.54 | 1.43× | 1.44 | 1.62× | 1.81 |
| | ✗ | TokenRecycling | 1.76× | 4.22 | 1.99× | 3.10 | 2.31× | 3.31 | 2.15× | 3.35 | 2.18× | 3.39 | 2.08× | 3.47 |
| | ✗ | FLy (Ours) | **2.02×** | **12.30** | **3.57×** | **14.85** | **2.76×** | **11.45** | **2.39×** | **11.02** | **2.96×** | **12.14** | **2.74×** | **12.41** |
| L31 405B | ✗ | SpS | 2.32× | 14.02 | 3.72× | 24.13 | 2.47× | 11.01 | 2.63× | 11.22 | 3.01× | 12.51 | 2.83× | 14.58 |
| | ✗ | REST | 1.94× | 2.09 | 2.77× | 3.02 | 1.40× | 1.46 | 1.43× | 1.53 | 1.79× | 1.44 | 1.79× | 1.91 |
| | ✗ | TokenRecycling | 1.53× | 3.52 | 1.60× | 3.12 | 1.72× | 3.21 | 1.79× | 3.34 | 1.78× | 3.35 | 1.68× | 3.31 |
| | ✗ | FLy (Ours) | **2.72×** | **15.08** | **4.07×** | **24.15** | **5.16×** | **14.49** | **5.67×** | **15.75** | **6.37×** | **16.18** | **4.80×** | **17.13** |
| | | | | | | | Temperature=1 | | | | | | | |
| L33 70B | ✓ | EAGLE-3 | 1.43× | 3.60 | 1.99× | 2.27 | 1.47× | 2.05 | 1.58× | 2.45 | 0.81× | 1.18 | 1.45× | 2.31 |
| | ✗ | FLy (Ours) | **1.89×** | **11.02** | **3.37×** | **13.78** | **2.72×** | **11.97** | **2.21×** | **10.99** | **2.61×** | **11.01** | **2.56×** | **11.75** |
| L31 70B | ✗ | SpS | 1.71× | 11.28 | 3.33× | 15.73 | 1.79× | 9.14 | 1.59× | 8.64 | 1.58× | 8.74 | 2.00× | 10.71 |
| | ✗ | REST | 1.29× | 1.50 | 2.51× | 2.89 | 1.37× | 1.40 | 1.22× | 1.47 | 1.36× | 1.38 | 1.55× | 1.73 |
| | ✗ | TokenRecycling | **1.83×** | 4.38 | 2.02× | 3.15 | 2.30× | 3.33 | 2.17× | 3.36 | 2.17× | 3.38 | 2.10× | 3.52 |
| | ✗ | FLy (Ours) | 1.81× | **11.69** | **3.64×** | **15.83** | **2.56×** | **11.77** | **2.35×** | **11.27** | **2.71×** | **11.63** | **2.62×** | **12.44** |
| L31 405B | ✗ | SpS | 1.97× | 11.70 | 3.13× | 20.15 | 2.10× | 9.19 | 2.20× | 9.38 | 2.55× | 10.45 | 2.39× | 12.17 |
| | ✗ | REST | 1.81× | 1.95 | 2.58× | 2.82 | 1.31× | 1.36 | 1.33× | 1.43 | 1.30× | 1.34 | 1.67× | 1.78 |
| | ✗ | TokenRecycling | 1.53× | 3.54 | 1.79× | 3.10 | 1.73× | 3.22 | 1.79× | 3.34 | 1.78× | 3.34 | 1.68× | 3.31 |
| | ✗ | FLy (Ours) | **2.74×** | **15.44** | **4.10×** | **24.09** | **6.08×** | **17.97** | **6.42×** | **18.84** | **6.71×** | **18.10** | **5.21×** | **18.89** |

Table 2: Speedup ratios and mean accepted tokens ($\tau$) on out-of-domain (OOD) datasets. L31 and L33 represents Llama-3.1-Instruct and Llama-3.3-Instruct, respectively. Mean represents the average performance across these datasets. We use **bold text** to denote the best result. ✓ indicates training-based methods, whereas ✗ means training-free methods.

for maintaining the target model's performance, we conduct an accuracy preservation benchmark in Section 3.3. Finally, we perform a series of ablation studies to validate the effectiveness of individual components of our method in Section 3.4.

## 3.1 EXPERIMENTAL SETUP

**Models.** For training-free baselines, we employ Llama-3.1-8B-Instruct (Dubey et al., 2024) as the draft model, with Llama-3.1-70B-Instruct and Llama-3.1-405B-Instruct as target models. For comparisons with EAGLE-2, we use Meta-Llama-3-Instruct-70B as the target model and Meta-Llama-3-Instruct-8B as its draft model. For comparisons with EAGLE-3, we adopt Llama-3.3-70B-Instruct as the target model and Llama-3.1-8B-Instruct as its draft model.

**Baselines.** We compare our method against both training-based and training-free SPD approaches. For training-based methods, we primarily include the previous state-of-the-art EAGLE-2 (Li et al., 2024b) and EAGLE-3 (Li et al., 2025b). For training-free methods, we select Speculative Sampling (Xia et al., 2023) (SpS), REST (He et al., 2023) and TokenRecycling (Luo et al., 2024) as representative baselines.

**Benchmarks.** To assess robustness under distribution shift, we divide the evaluation datasets into out-of-domain (OOD) and in-domain (ID) categories according to the training data used for EAGLE-3. Specifically, the OOD group includes ACP-prog-gen (Kokel et al., 2025) (Action, Change, and Planning) which focuses on planning-style reasoning, NIAH-multivalue (Hsieh et al., 2024) which extends the classic Needle-in-a-Haystack test to include varied needle types and counts, and MGSM (Shi et al., 2022), the Multilingual Grade School Math benchmark. For the ID group, we adopt the widely used GSM8K (Cobbe et al., 2021), HumanEval (Chen et al., 2021), and MBPP (Austin et al., 2021) benchmarks.

**Implementation Details.** We set the deferred window length to $W = 6$ and the entropy gate threshold to $\theta = 0.3$ for all experiments. The draft token number $K$ at each round is set to 15 for the 70B target model and 25 for the 405B target model. All experiments are conducted on AMD Instinct MI355X GPUs. Specifically, the 70B model is evaluated on a single GPU, whereas the 405B model is distributed across four GPUs.

## 3.2 PERFORMANCE COMPARISON

On OOD datasets, as shown in Table 2, FLy demonstrates exceptional performance. For Llama-3.1-70B-Instruct, FLy achieves an average speedup of 2.74× (2.62×) with temperature= 0 (= 1), outperforming existing training-free baselines. On the Llama-3.3 variant, FLy surpasses the training-

| Model | Training | Method | GSM8K | | HumanEval | | MBPP | | Mean | |
|---|---|---|---|---|---|---|---|---|---|---|
| | | | Speedup | $\tau$ | Speedup | $\tau$ | Speedup | $\tau$ | Speedup | $\tau$ |
| | | | | | Temperature=0 | | | | | |
| L3 70B | ✓ | EAGLE-2 | 2.54× | 3.74 | 2.56× | 4.24 | **2.61×** | 3.90 | 2.57× | 3.96 |
| | ✗ | FLy (Ours) | **2.94×** | **11.67** | **2.60×** | **12.74** | 2.53× | **11.21** | **2.69×** | **11.87** |
| L33 70B | ✓ | EAGLE-3 | **3.72×** | 5.64 | **3.97×** | 5.73 | **3.80×** | 5.47 | **3.83×** | 5.61 |
| | ✗ | FLy (Ours) | 2.68× | **11.65** | 2.75× | **12.20** | 2.47× | **11.76** | 2.63× | **11.87** |
| L31 70B | ✗ | SpS | 2.13× | 9.87 | 1.64× | 9.89 | 1.72× | 10.47 | 1.83× | 10.08 |
| | ✗ | REST | 1.79× | 2.00 | 1.89× | 2.22 | 2.03× | 2.34 | 1.90× | 2.19 |
| | ✗ | TokenRecycling | 2.10× | 3.04 | 2.12× | 3.07 | 2.13× | 3.01 | 2.12× | 3.04 |
| | ✗ | FLy (Ours) | **2.98×** | **12.57** | **2.86×** | **12.61** | **2.79×** | **12.84** | **2.88×** | **12.67** |
| L31 405B | ✗ | SpS | 2.59× | 10.98 | 2.84× | 11.62 | 3.03× | 12.67 | 2.82× | 11.76 |
| | ✗ | REST | 2.07× | 2.13 | 2.14× | 2.41 | 2.36× | 2.46 | 2.19× | 2.33 |
| | ✗ | TokenRecycling | 1.47× | 2.83 | 1.54× | 2.88 | 1.66× | 2.92 | 1.56× | 2.88 |
| | ✗ | FLy (Ours) | **4.61×** | **17.02** | **5.15×** | **15.83** | **6.26×** | **18.56** | **5.34×** | **17.14** |
| | | | | | Temperature=1 | | | | | |
| L3 70B | ✓ | EAGLE-2 | 2.36× | 3.48 | 2.38× | 3.95 | 2.43× | 3.63 | 2.39× | 3.69 |
| | ✗ | FLy (Ours) | **2.91×** | **11.55** | **2.53×** | **12.67** | **2.59×** | **11.37** | **2.68×** | **11.86** |
| L33 70B | ✓ | EAGLE-3 | **3.50×** | 5.31 | **3.74×** | 5.40 | **3.58×** | 5.15 | **3.61×** | 5.29 |
| | ✗ | FLy (Ours) | 2.85× | **12.40** | 2.75× | **12.55** | 2.69× | **12.33** | 2.76× | **12.43** |
| L31 70B | ✗ | SpS | 2.13× | 9.70 | 1.07× | 6.29 | 1.00× | 6.08 | 1.40× | 7.36 |
| | ✗ | REST | 1.71× | 1.91 | 1.81× | 2.13 | 1.94× | 2.24 | 1.82× | 2.09 |
| | ✗ | TokenRecycling | 2.08× | 3.02 | 2.15× | 3.06 | 2.15× | 3.01 | 2.13× | 3.03 |
| | ✗ | FLy (Ours) | **2.67×** | **11.93** | **2.52×** | **11.96** | **2.59×** | **12.57** | **2.59×** | **12.15** |
| L31 405B | ✗ | SpS | 2.18× | 9.25 | 2.39× | 9.78 | 2.55× | 10.67 | 2.37× | 9.90 |
| | ✗ | REST | 1.78× | 1.82 | 1.83× | 2.07 | 2.02× | 2.11 | 1.88× | 2.00 |
| | ✗ | TokenRecycling | 1.48× | 2.85 | 1.56× | 2.91 | 1.67× | 2.92 | 1.57× | 2.89 |
| | ✗ | FLy (Ours) | **4.97×** | **19.40** | **5.00×** | **16.29** | **5.97×** | **17.80** | **5.31×** | **17.83** |

Table 3: Speedup ratios and mean accepted tokens ($\tau$) on in-domain (ID) datasets. L3, L31 and L33 represents Meta-Llama-3-Instruct, Llama-3.1-Instruct and Llama-3.3-Instruct, respectively. Mean represents the average performance across these datasets. We use **bold text** to denote the best result. ✓ indicates training-based methods, whereas ✗ means training-free methods.

based SOTA method EAGLE-3 by 1.62× (1.77×). This advantage scales with model size. FLy achieves a 4.80× (5.21×) average speedup on the 405B variant, as its higher per-token latency allows greater time savings when draft tokens are accepted, reducing costly target model calls.

On ID datasets, presented in Table 3, FLy remains highly competitive. With the 70B model, FLy obtains a 2.88× (2.59×) average speedup, surpassing all other training-free methods. Notably, FLy outperforms the training-based EAGLE-2 by 1.05× (1.12×). Although a bit slower than the heavily optimized EAGLE-3, this is expected and reasonable for a plug-and-play, training-free approach. On the larger 405B model, FLy achieves a remarkable 5.34× (5.31×) speedup, significantly exceeding all baselines and highlighting its powerful potential for accelerating large-scale models. It is particularly noteworthy that training-based methods like EAGLE-3 require prohibitive resource costs to train a drafter for the 405B model, often resulting in no officially supported model at this scale. In contrast, our training-free approach provides a solution that delivers excellent performance without such overhead.

Experimental results demonstrate that FLy is more robust under distribution shift. EAGLE-3 achieves an average speed-up of 3.83× on in-distribution (ID) datasets, but this drops to 1.56× on out-of-distribution (OOD) data. In contrast, when using Llama-3.3-70B as the target model, FLy attains an average acceleration of 2.63× on ID data and 2.53× on OOD data, indicating that its performance remains far more stable across the distribution shift.

## 3.3 ACCURACY PRESERVATION

Since our method is a loosely SPD method, the output of the accelerated target model would not be exactly the same as the original. Thus, we report accuracy preservation relative to the original target model after applying FLy, as shown in Figure.3. Concretely, we normalize the original target model's score to 100% and use a recovery ratio to quantify how much performance is retained. The original non-normalized scores are provided in Appendix B. Across different datasets and model scales, our method consistently maintains accuracy with over 99% recovery score, and performs on par with the training-based loosely SPD method JudgeDecoding (Bachmann et al., 2025). The JudgeDecoding accuracy preservation results reported here are taken directly from the original paper, since JudgeDecoding has not been open-sourced. It is worth noting that JudgeDecoding also recognizes sensitivity to train–test domain misalignment. When coding examples are removed from their training data, performance on HumanEval drops substantially from 99.4% to 92.3%, underscoring degradation under distribution shift.

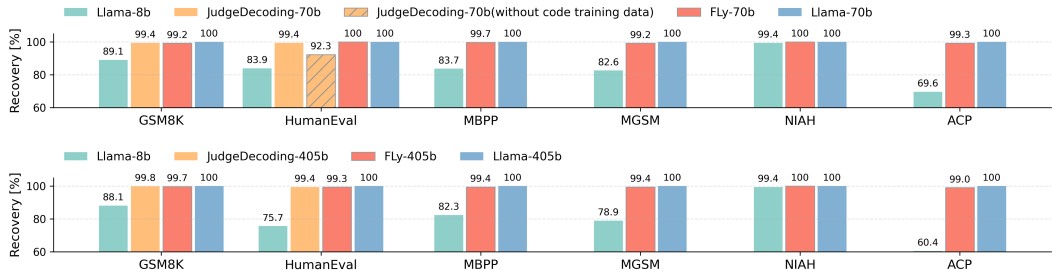

Figure 3: Accuracy preservation results. The performance of the target model is normalized to 100, and the recovery ratio is used to quantify the extent of performance preservation.

## 3.4 ABLATION STUDIES

In this section, we evaluate the contribution of each component of our method to the overall performance. Specifically, we conduct ablation studies on four key aspects: (1) the core hyper-parameters, including window length ($W$), the entropy gate threshold ($\theta$), and the draft token number ($K$). (2) The multi-level acceleration mechanism, which evaluates the impact of accelerating the drafter stage itself. (3) The drafter model size, capturing the capacity–latency trade-off. (4) Cross-model draft–target combinations, evaluating robustness across LLM families and scales. Unless otherwise specified, experiments are conducted with Llama-3.1-70B-Instruct as the target LLM, Llama-3.1-8B-Instruct as the drafter, HumanEval as the evaluation dataset, and the temperature fixed as 0.

**Hyper-parameters.** The window length ($W$) determines how many subsequent positions are monitored in the deferred decision process to assess the target model's behavior. We vary $W \in \{0, 4, 6, 8\}$, as shown in Table 4 (1). When the window is disabled ($W = 0$), all deferred mismatches are accepted, leading to maximal speedup and $\tau$, but at the expense of accuracy. A short window ($W = 4$) yields a higher speedup with a slight accuracy drop. $W = 6$ restores perfect recovery with only a small loss in speedup and is our default, while $W = 8$ is overly strict, reducing both $\tau$ and end-to-end speedup.

The entropy gate threshold ($\theta$) controls whether a position is treated as a unique choice (low entropy, mismatches directly rejected) or multiple plausible candidates (high entropy, deferral allowed). We study $\theta \in \{0, 0.3, 0.6, 1\}$, as shown in Table 4 (2). Disabling the gate ($\theta = 0$) defers all mismatches, maximizing speedup and $\tau$ but hurting accuracy. $\theta = 0.3$ achieves perfect recovery with strong speedup and is our default. A stricter gate ($\theta = 0.6$) reduces deferrals and rejects most mismatches, lowering speedup and $\tau$. When $\theta = 1$, the method rejects all mismatches and degenerates to standard SPD.

| Hyper-parameter | Val | Speedup | $\tau$ | Recovery(%) |
|---|---|---|---|---|
| (1) Window length $W$ | 0 | 3.42× | 15.59 | 93.7 |
| | 4 | 2.91× | 13.27 | 97.9 |
| | 6 | 2.86× | 12.61 | 100 |
| | 8 | 2.61× | 11.96 | 100 |
| (2) Entropy threshold $\theta$ | 0 | 2.98× | 12.76 | 97.7 |
| | 0.3 | 2.86× | 12.61 | 100 |
| | 0.6 | 2.76× | 12.20 | 100 |
| | 1 | 1.64× | 9.89 | 100 |
| (3) Draft token $K$ | 10 | 2.75× | 9.04 | 100 |
| | 15 | 2.86× | 12.61 | 100 |
| | 20 | 2.59× | 15.60 | 100 |
| | 25 | 2.58× | 18.56 | 100 |

Table 4: Ablation study on the hyper-parameters.

The draft token number $K$ specifies how many tokens the drafter generates in a single round. We evaluate $K \in \{10, 15, 20, 25\}$, as shown in Table 4 (3). While $\tau$ grows monotonically with $K$, speedup increases then declines as the drafter's cost for unaccepted tokens ($K - \tau$) becomes the bottleneck. We therefore use $K = 15$ for the 70B target (best speedup) and $K = 25$ for the 405B target, where verification is costlier. A detailed 405B ablation is shown in Appendix C.

**Multi-level acceleration effect.** Ablation results on multi-level acceleration are presented in Table 5. Enabling MLA improves speedup from 2.69× to 2.86×, confirming that accelerating the drafter stage reduces end-to-end latency beyond what target-only acceleration can achieve.

**Draft model size.** We vary the draft model size from 1B to 8B, as shown in Table 6. The mean accepted tokens $\tau$ grows monotonically with drafter capacity, whereas the overall speedup is non-monotonic since the drafter's computational cost also increases with model size, potentially out-

weighing the benefit of a higher $\tau$ unless the gain from $\tau$ is substantial. We therefore adopt the 8B model as the default drafter, as it achieves the best balance between accuracy and throughput.

|  | w/o MLA | w/ MLA |
| --- | --- | --- |
| Speedup | 2.69× | 2.86× |

Table 5: Ablation study on the multi-level acceleration.

| Draft model size | 1B | 3B | 8B |
| --- | --- | --- | --- |
| Speedup | 2.80× | 2.47× | 2.86× |
| $\tau$ | 8.98 | 10.71 | 12.61 |
| Recovery(%) | 99.23 | 100 | 100 |

Table 6: Ablation study on drafter size.

**Cross-model draft–target pairing.** To validate whether FLy generalizes beyond a specific pair of models, we further evaluate it under three heterogeneous draft–target combinations, as shown in Table 7. They cover both cross-family and cross-scale settings: (1) a lightweight Qwen2.5-Coder-0.5B-Instruct (Hui et al., 2024; Yang et al., 2024) drafter with a Mistral-Large-Instruct-2411 target, (2) a DeepSeek-R1-Distill-Qwen-7B (DeepSeek-AI, 2025) drafter with a Mistral-Large-Instruct-2411 target, and (3) a DeepSeek-R1-Distill-Qwen-1.5B drafter accelerating a larger Llama variant. Across all settings, FLy achieves 1.85× to 3.54× speedups while preserving over 99% of the target model's accuracy. This demonstrates FLy's model-agnostic property, which allows it to seamlessly compose with multiple draft–target pairs without any retraining.

| Draft model | Target model | Speedup | $\tau$ | Recovery (%) |
| --- | --- | --- | --- | --- |
| Qwen2.5-Coder-0.5B-Instruct | Mistral-Large-Instruct-2411 | 3.54× | 12.34 | 99.1 |
| DeepSeek-R1-Distill-Qwen-7B | Mistral-Large-Instruct-2411 | 2.35× | 9.60 | 100 |
| DeepSeek-R1-Distill-Qwen-1.5B | DeepSeek-R1-Distill-Llama-70B | 1.85× | 10.28 | 100 |

Table 7: Ablation study on cross-model draft–target pairing.

## 4 RELATED WORKS

Speculative decoding (SPD) speeds up autoregressive LLMs by letting a small drafter propose several next tokens sequentially and having the target model verify them in parallel (Xia et al., 2023).

**Training-based SPD.** A large body of research (Li et al., 2025a; Liu et al., 2024; Xiao et al., 2024; Ankner et al., 2024) extends SPD by learning auxiliary predictors to better align drafter and target, thereby improving speculation accuracy. Typical strategies train additional modules that imitate the target's behavior to raise acceptance rates and speedups. Medusa (Cai et al., 2024) uses hidden states from the base LLM as inputs to multiple lightweight MLP heads, each predicting a future token. GLIDE (Du et al., 2024) speeds up decoding by reusing the target model's KV cache via cross-attention and adaptively expanding proposals. HASS (Zhang et al., 2024) improves acceptance rates by aligning training objectives with inference behavior. EAGLE (Li et al., 2024a) generalizes this design by employing lightweight Transformer predictors and concatenated token–state pairs, while EAGLE-2 (Li et al., 2024b) improves efficiency with a dynamic tree–based candidate selection mechanism. EAGLE-3 (Li et al., 2025b) further leverages intermediate hidden states to scale up decoding acceleration. Distinct from attaching auxiliary modules, LayerSkip (Elhoushi et al., 2024) exploits the target itself by training early layers to propose drafts that are subsequently verified by the full model. Although effective, these approaches require task-specific supervision and often generalize poorly beyond the training distribution, limiting robustness in out-of-distribution settings.

**Training-free SPD.** In contrast, training-free approaches dispense with additional training. Speculative Sampling (Chen et al., 2023) pioneered the foundational draft-then-verify paradigm, employing a modified rejection sampling scheme to accelerate decoding while ensuring lossless generation. Retrieval-based methods (Zhao et al., 2024; He et al., 2023) maintain a datastore and retrieve $n$-gram continuations as draft tokens, offering a plug-and-play accelerator that obviates a parametric drafter and can be applied to arbitrary targets. Draft&Verify (**?**) formalizes this by employing Bayesian optimization to identify optimal static skipped layer sets. SWIFT (**?**) further advances this by dynamically optimizing the skipped layer set on-the-fly based on the input context. Additionally, KNN-SSD (**?**) utilizes nearest neighbor search to retrieve domain-specific skipping configurations

to address sensitivity to data distribution shifts. Because the drafts are generally weaker and less aligned with the target, these approaches tend to yield lower speedups than training-based ones.

**Loosely SPD.** Strict verification often rejects *semantically valid* drafts, limiting achievable speedups. To address this, "loosely" variants (Garipov et al., 2025; Wang et al., 2025c) relax the criterion to tolerate token-level deviations. JudgeDecoding (Bachmann et al., 2025) trains a lightweight auxiliary model to recognize semantically correct but mismatched drafts, but such supervised training limits out-of-domain generalization. Reflective Verification (Wang et al., 2025b) takes a training-free route, leveraging the reflective capacity of LLMs to semantically probe draft correctness. SPRINTER (Zhong et al., 2025) employs a lightweight verifier to approximate target acceptance only when mismatches occur, reducing target calls and further lowering latency. Alignment-sampling methods (Wang et al., 2025a) exploit distributional information from the prefilling phase to propose better-aligned drafts, and couple this with flexible thresholding strategies that adaptively accept high-quality but imperfect candidates. These approaches highlight the growing interest in relaxing strict token-level equivalence, trading exactness for efficiency while exploring balances between training-based and training-free designs.

Our FLy follows the loose philosophy but differs from prior methods: instead of relying on trained verifiers or relaxed acceptance rules, it remains entirely training-free and judges drafts by deferring verification, leveraging the target's own behavior and entropy signals to distinguish harmless divergences from true errors.

## 5 LIMITATIONS

Although FLy achieves substantial speedups while preserving high accuracy, it remains fully training-free and does not optimize its verification policy on task-specific benchmarks. On datasets tailored to training-based speculative decoding methods such as EAGLE-3, this can lead to a performance gap. Moreover, FLy is designed to preserve semantic consistency with the target model rather than exact token-level agreement, which can be suboptimal for applications that require verbatim reproduction (e.g., repeating a long passage or reciting a poem).

## 6 CONCLUSION

This paper introduces Training-Free Loosely Speculative Decoding (FLy), a novel training-free algorithm that replaces standard SPD's rigid exact-match criterion with a loosely verified scheme to accept semantically correct tokens. When a mismatch occurs, FLy applies a two-tier scheme to distinguish genuine errors from semantically valid cases. Firstly, an **entropy-level gate** determines whether loosely verifying is appropriate (*i.e.*, whether the current context is sufficiently uncertain such that multiple alternative tokens could be considered valid). Then, a **token-level deferred window** monitors the target model's behavior over the subsequent tokens. If the generation proceeds without further divergence, the initial mismatch is treated as a *differently worded yet semantically correct* continuation and is accepted. Otherwise, additional mismatches indicate corrective behavior, and the initial mismatch is retroactively rejected. We further propose a **multi-level acceleration** mechanism, in which the drafter itself is accelerated to further reduce latency. Experimental results show that our approach significantly increases mean accepted tokens ($\tau$) and the overall speedup while preserving the target model's accuracy ($\geq 99\%$), while demonstrating strong generalization.

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

# A  DETAILED ALGORITHM

We provide a step-by-step exposition of our proposed FLy in Section 2.2, detailing each component and procedure to facilitate deeper understanding.

---

**Algorithm 1:** Training-$\underline{\text{F}}$ree $\underline{\text{L}}$oosel$\underline{\text{y}}$ Speculative Decoding (FLy)

---

**Input:** Prompt $x$, draft model $\mathcal{M}_D$, target model $\mathcal{M}_T$, draft token number $K$, thresholds $\theta$, window size $W$, and vocabulary $\mathcal{V}$.

**Output:** Generated sequence $y$, mean accepted token $\tau$.

```
// Initialize SPD round t, accepted token number s
```

1   $t \leftarrow 0, \ s \leftarrow 0$

2   $y \leftarrow x$

3   **while** *not EOS* **do**

     `// (Step 1) Sequentially draft K tokens`

4      $\{\hat{y}_i^t\}_{i=1}^K \leftarrow \mathcal{M}_D(y; K)$

     `// (Step 2) Verify by target model with a single forward pass`

5      $\{\ell_i^t\}_{i=1}^{K+1} \leftarrow \mathcal{M}_T.\text{forward}(\text{concat}\,[y_{-1}^{t-1}, \{\hat{y}_i^t\}_{i=1}^K])$

6      **for** $i \leftarrow 1$ **to** $K$ **do**

7          $p_{\mathcal{M}_T,i}(v) \leftarrow \text{softmax}(\ell_i^t)_v \text{ for } v \in \mathcal{V} \ , \ y_i^t \leftarrow \arg\max_{v \in \mathcal{V}} p_{\mathcal{M}_T,i}(v)$

8          $\Delta_i^t \leftarrow \mathbb{1}[\hat{y}_i^t = y_i^t]$

9      $p_{\mathcal{M}_T,K+1}(v) \leftarrow \text{softmax}(\ell_{K+1}^t)_v \text{ for } v \in \mathcal{V} \ , \ y_{K+1}^t \leftarrow \arg\max_{v \in \mathcal{V}} p_{\mathcal{M}_T,K+1}(v)$

10      $J \leftarrow \{\, i \mid \Delta_i^t = 0 \,\}$

11      **if** $J = \phi$ **then**

         `// All matched, accept all`

12          $y \leftarrow \text{concat}\,[y, \{\hat{y}_i^t\}_{i=1}^K, y_{K+1}^t]$

13          $t \leftarrow t + 1$

14          $s \leftarrow s + K + 1$

15          **continue**

16      $\texttt{accept\_all} \leftarrow \texttt{True}$

17      **for** $j \in J$ **do**

         `// (Step 3) Entropy gate at the mismatch j`

18          $h_j \leftarrow \dfrac{-\sum_{v \in \mathcal{V}} p_{\mathcal{M}_T,j}(v) \log p_{\mathcal{M}_T,j}(v)}{\log |\mathcal{V}|}$

19          **if** $h_j < \theta$ **then**

            `// Strict mode, reject`

20             $\texttt{accept\_all} \leftarrow \texttt{False}$

21             $y \leftarrow \text{concat}\,[y, \{\hat{y}_i^t\}_{i=1}^{j-1}, \ y_j^t]$

22             $s \leftarrow s + j$

23             **break**

24          **else**

            `// (Step 4) Deferred window decision over next W tokens`

25             $M_W \leftarrow \sum_{i=j+1}^{j+W}(1 - \Delta_i^t)$

26             **if** $M_W = 0$ *and* $j + W \leq K$ **then**

               `// Mismatch at j is worded-differently but semantically`
                  `correct, accept it`

27                **continue**

28             **else**

               `// Treat it as harmful, reject it`

29                $\texttt{accept\_all} \leftarrow \texttt{False}$

30                $y \leftarrow \text{concat}\,[y, \{\hat{y}_i^t\}_{i=1}^{j-1}, \ y_j^t]$

31                $s \leftarrow s + j$

32                **break**

33      **if** $\texttt{accept\_all}$ **then**

34          $y \leftarrow \text{concat}\,[y, \{\hat{y}_i^t\}_{i=1}^K, y_{K+1}^t]$

35          $s \leftarrow s + K + 1$

36      $t \leftarrow t + 1$

37   $\tau \leftarrow s/t$

---

## B  RAW ACCURACY SCORES

This section presents the raw, non-normalized accuracy scores corresponding to the accuracy preservation results discussed in Section 3.3. Table 8 provides the original evaluation metrics for each model and dataset, offering a direct view of the absolute performance.

| Model Family | Model | GSM8K | HumanEval | MBPP | MGSM | NIAH | ACP |
|---|---|---|---|---|---|---|---|
| | L31 8b | 85.06 | 66.46 | 74.87 | 73.20 | 99.42 | 42.30 |
| | FLy 70b | 94.69 | 79.26 | 89.15 | 88.41 | 99.93 | 60.37 |
| Llama-3.1-Instruct | L31 70b | 95.45 | 79.26 | 89.42 | 88.67 | 99.97 | 60.77 |
| | FLy 405b | 96.21 | 87.20 | 90.48 | 92.27 | 100 | 69.31 |
| | L31 405b | 96.51 | 87.80 | 91.00 | 92.80 | 100 | 70.00 |
| Llama-3.3-Instruct | FLy 70b | 94.84 | 84.54 | 88.62 | 90.2 | 100 | 59.23 |
| | L33 70b | 95.60 | 85.37 | 89.15 | 90.4 | 100 | 57.69 |
| | L3 8b | 79.30 | 60.37 | 75.40 | / | / | / |
| Meta-Llama-3-Instruct | FLy 70b | 91.75 | 78.55 | 84.39 | / | / | / |
| | L3 70b | 92.34 | 79.27 | 84.93 | / | / | / |

Table 8: Raw accuracy scores. L31 represents Llama-3.1-Instruct, L33 represents Llama-3.3-Instruct, L3 represents Meta-Llama-3-Instruct and FLy is our proposed method.

## C  ABLATION STUDY ON DRAFT TOKEN NUMBER $K$ FOR THE 405B VARIANT

We investigate the effect of the draft token number $K$ on the performance of FLy when accelerating the Llama-3.1-405B-Instruct model. As shown in Table 9, increasing $K$ leads to a higher mean accepted token count ($\tau$) and improved speedup, but at the cost of slightly reduced accuracy recovery. Based on the trade-off, we select $K = 25$ for the 405B model, which achieves a strong speedup of $5.15\times$ with minimal accuracy degradation (99.3% recovery).

| Draft token $K$ | Speedup | $\tau$ | Recovery(%) |
|---|---|---|---|
| 15 | 4.60× | 11.43 | 99.7% |
| 20 | 4.89× | 13.64 | 99.7% |
| 25 | 5.15× | 15.83 | 99.3% |
| 30 | 5.67× | 17.57 | 97.9% |

Table 9: Ablation study on draft token number $K$ for Llama-3.1-405b-Instruct.

## D  SPEEDUP ON LARGE BATCH SIZES BASED ON vLLM

We evaluate the impact of FLy on throughput for large batch sizes using four AMD Instruct MI250 GPUs with vLLM, a widely adopted production-grade framework. The results are shown in Table 10.

| Batch size | 2 | 4 | 8 | 16 |
|---|---|---|---|---|
| Llama-3.1-70b-Instruct | 2.39× | 2.09× | 1.96× | 1.78× |

Table 10: Throughputs on HumanEval dataset under different batch sizes using vLLM. The inference speed of the model without speculative sampling is used as the baseline (1.00×).

## E  CASE STUDY

Figure 4 presents a case from the GSM8K dataset using Llama-3.1-405B-Instruct, where FLy correctly accepts semantically valid draft tokens that standard SPD would reject. As shown, disagreements between the drafter and target appear at several tokens. Rather than rejecting the draft whole-

sale as standard SPD would, FLy preserves semantically valid alternatives, increasing acceptance rate per step and accelerating decoding.

| **Input Prompt:** |
| --- |
| <\|start_header_id\|>user<\|end_header_id\|>
Given the following problem, reason and give a final answer to the problem.
Problem: Ellen decided to play a prank on her friend. She got a case of 12 sodas and shook 3 of them up. Then she took 1 unshaken soda for herself and left. Ellen's brother stopped by and took 1 of the shaken sodas and 2 of the unshaken sodas, then Ellen's friend came along. What is the likelihood, expressed as a percentage, that Ellen's friend gets sprayed with soda from a shaken can?
Your response should end with "The final answer is [answer]" where [answer] is the response to the problem.<\|eot_id\|><\|start_header_id\|>assistant<\|end_header_id\|> |

| **Output:** |
| --- |
| Ellen (There) started with 12 sodas. She shook 3 of them. Then (So) she took 1 unshaken soda, ~~leaving~~ so there ~~were~~ are 12 - 1 = 11 sodas left. Her brother took 1 shaken soda and 2 unshaken sodas (ones), so there are 11 - 3 = 8 sodas left. There were originally (3) 3 shaken sodas, but 1 was taken ~~by~~ , so there are 2 shaken sodas left. ~~Ell~~ The chance of Ellen's friend getting sprayed (a) is the number of shaken sodas divided by the total number of sodas. There are 2 shaken sodas out (and) of 8 total sodas. 2 / 8 = 0.25. To express (convert) this as a percentage, multiply by 100. 0.25 * 100 = 25%. The final answer is 25%. <\|eot_id\|><\|start_header_id\|>assistant<\|end_header_id\|> |

Figure 4: Case study of FLy on a sample from the GSM8K dataset using Llama-3.1-405B-Instruct. Blue tokens denote the drafter's output and green tokens denote the target's output. Under standard speculative decoding, all mismatches would be rejected. In contrast, FLy selectively rejects only the mismatches shown with ~~strikethrough~~, while retaining the remaining ones because they are semantically valid.

# F  LLM USAGE

In the preparation of this manuscript, we used a large language model (LLM) solely for language polishing and grammatical refinement of the text.

