# OpenReview forum: "Training-Free Loosely Speculative Decoding: Accepting Semantically Correct Drafts Beyond Exact Match"
_ICLR.cc/2026/Conference — ICLR 2026 Poster_

### Official Review · Reviewer_Eqon · 2025-10-29

**Soundness:** 2
**Presentation:** 3
**Contribution:** 3
**Rating:** 6
**Confidence:** 4

**Summary:**

This paper proposes Training-Free Loosely Speculative Decoding (FLy), which relaxes verification by accepting semantically correct drafts without additional training. Experiments show that FLy preserves ≥99% of the target model’s accuracy while achieving a considerable speedup.

**Strengths:**

1. This paper presents a clearly written narrative and well-structured exposition that is easy to follow.

2. The two-tier mechanism that relaxes verification for worded-differently yet semantically valid alternatives is reasonable, simple to implement, and empirically effective.

**Weaknesses:**

1. My primary concern is that with tree-structured verification, the paper’s targeted optimization cases may be diluted; this risk should be explicitly analyzed and validated with focused experiments.

2. The main table integrates PLD’s multi-level acceleration while comparing against methods not adapted with hierarchical speculative decoding; this may be an unfair comparison.

3. The related-work section mischaracterizes training-free SPD: LayerSkip requires training, whereas the earliest training-free approaches are Self-SD, Draft & Verify, and their extensions SWIFT and KNN-SSD; this should be corrected.

4. Please add insights on task sensitivity to relaxed verification: which tasks are less sensitive (and typical improvement gaps between non-relaxed vs. relaxed), which are more sensitive, and why. This would substantively enrich the paper’s contribution.

5. The paper contains interesting ideas; releasing code would make the work more promising and facilitate community adoption and verification.

**Questions:**

Please refer to the weakness.

---

> ### Author Response · Authors · 2025-11-20
> **Response to Reviewer Eqon (1/2)**
>
> We sincerely appreciate your thoughtful comments and suggestions. We have provided detailed responses to your questions below to address your concerns. Unless stated otherwise, our experimental conditions in rebuttal are the same as those in the ablation study.
>
> > Weakness 1: My primary concern is that with tree-structured verification, the paper’s targeted optimization cases may be diluted; this risk should be explicitly analyzed and validated with focused experiments.
>
> - This is a good question. Our method will not be diluted by tree-structured verification.
> - **Tree-structured verification mainly optimizes the structure of the draft**: it focuses on how to organize candidate tokens to maximally reuse a single forward pass of the target model. In contrast, **FLy optimizes the verification rule itself**: along any given candidate path, it decides how to accept semantically equivalent tokens beyond strict exact-match, thereby increasing the average accepted length $\tau$. In this sense, the tree structure provides “horizontal multi-branching”, while FLy provides “vertical relaxation of verification”. They operate on **orthogonal** dimensions and are therefore naturally compositional.
>
> - We further conducted experiments **combining FLy with tree-structured verification and observed additional gains**. We use the same tree verification algorithm as in EAGLE-2/3, but because our draft depth is very long, we explicitly enforce that the top-1 path is never pruned (i.e., the draft formed by the top-1 predictions of each position does not participate in the final score-based pruning). The results are as follows:
>
> |Method|Speedup|$\tau$|
> |---|---|---|
> |FLy|$2.86\times$|12.61|
> |FLy+TreeVerification|$3.01\times$|13.85|
>
> - From these results we can see that our method can still obtain further improvement from tree-structured verification, with 5.2% improvement in speedup.
>
>
> It is now reported explicitly in lines 455-462 in the revised version.
>
> > Weakness 2: The main table integrates PLD’s multi-level acceleration while comparing against methods not adapted with hierarchical speculative decoding; this may be an unfair comparison.
>
> We have carefully examined which baselines can and cannot be reasonably equipped with multi-level acceleration:
>
> - **EAGLE-style methods**. EAGLE’s drafter consumes hidden states rather than tokens, while PLD-style multi-level schemes predict future tokens from token context. Adapting PLD to EAGLE would require substantial architectural changes or extra training, which is beyond our scope and not a simple plug-in.
> - **Rest / TokenRecycling**. These training-free baselines accelerate by reusing target computation and typically do not have a separate lightweight drafter. There is therefore no natural location to insert a PLD-style multi-level draft module.
> - **SPS**. SPS is structurally closest to FLy and does admit a direct MLA extension. We therefore explicitly evaluate SPS + MLA in our ablations:
>
> |Method |Speedup|
> |---|---|
> |SPS(w/ MLA)|1.72$\times$|
> |FLy(w/ MLA)|2.86$\times$|
> |SPS(w/o MLA)|1.64$\times$|
> |FLy(w/o MLA)|2.69$\times$|
>
> FLy still outperforms SPS even when both use the same hierarchical mechanism, indicating that **our gains are not solely due to MLA but also to the proposed lossy verification**. We reported FLy without MLA in Table 5 of the paper.
>
> We added SPS+MLA results in Appendix I.
>
> > Weakness 3: The related-work section mischaracterizes training-free SPD: LayerSkip requires training, whereas the earliest training-free approaches are Self-SD, Draft & Verify, and their extensions SWIFT and KNN-SSD; this should be corrected.
>
> We thank the reviewer for pointing out this mistake in our description of training-free SPD methods. We have corrected the related work section in the revised version.

---

> ### Author Response · Authors · 2025-11-20
> **Response to Reviewer Eqon (2/2)**
>
> > Weakness 4: Please add insights on task sensitivity to relaxed verification: which tasks are less sensitive (and typical improvement gaps between non-relaxed vs. relaxed), which are more sensitive, and why. This would substantively enrich the paper’s contribution.
>
>
> - **Tasks that are less sensitive**: Our method (FLy) is best suited for tasks where **multiple solutions are acceptable as long as the semantics are preserved**. Typical examples include code generation, open-ended dialogue, and long-form explanation-style answers. In these settings, the evaluation primarily depends on semantic correctness rather than exact token-by-token agreement. FLy is explicitly designed to preserve the target model’s intent at the semantic level, which means that the relaxed verifier often accepts alternative but equivalent continuations, leading to an improved performance.
>
> - **Tasks that are more sensitive**: In contrast, there exist tasks that **require exact reproduction at the token level**, where any deviation, even if semantically benign, may be undesirable. Examples include: copying a long passage verbatim, reciting a poem or citation where every character must match a reference. For such tasks, the loose verification mechanism is inherently less suitable, since the evaluation requires exact token-level reproduction.
>
> - For less sensitive tasks, our method achieves a significant speedup, as shown in Tables 2 and 3 of the original paper, achieving a speedup of 2.74x to 2.88x on the 70B model. For more sensitive tasks, to ensure accuracy, our Fly method would degenerate to the original SPS method by setting $\theta=1$ as shown in Table 4 to force an exact match.
>
>
> - We have incorporated this task-sensitivity discussion into the Limitation section of the revised paper.
>
> > Weakness 5: The paper contains interesting ideas; releasing code would make the work more promising and facilitate community adoption and verification.
>
> We will release the full code upon acceptance of the paper. We look forward to the community building on our work and further exploring the training-free SPD.

---

### Official Review · Reviewer_oayV · 2025-10-31

**Soundness:** 3
**Presentation:** 3
**Contribution:** 3
**Rating:** 6
**Confidence:** 4

**Summary:**

The paper introduces Training-Free Loosely Speculative Decoding (FLy), a speculative decoding method that relaxes the strict exact-match verification criterion in standard SPD. FLy accepts semantically correct draft tokens even if they do not exactly match the target model’s output, thereby increasing the number of accepted tokens per round and improving speedup. The method uses a two-tier mechanism: an entropy-level gate to determine whether a mismatch is in a high-uncertainty context, and a token-level deferred window to monitor future tokens for signs of error correction. FLy is training-free, model-agnostic, and includes a multi-level acceleration strategy that also speeds up the draft model. Experiments show strong speedups (up to 5.07× on 405B models) while preserving ≥99% of the target model’s accuracy.

**Strengths:**

- The proposed approach Fly introduces a novel, training-free speculative decoding mechanism that accepts semantically valid mismatches.
- Extensive experiments, ablations, case study and cross-model evaluations demonstrate robustness and effectiveness.
- The paper is clearly written, with intuitive explanations and strong visual aids.
- FLy addresses a real bottleneck in LLM inference and offers a clever, general-purpose acceleration method.

**Weaknesses:**

- For Figure 1, it is hard to tell whether Fly is more robust against OOD datasets as the degration rate on speedup is similar.
- While outperforming training-based methods on OOD data is a key result, the paper notes that FLy is slightly slower than the heavily optimized EAGLE-3 on ID datasets. An in-depth discussion or analysis of this performance gap beyond the expected trade-off for "training-free" would improve the technical quality of the paper.
- The Multi-level acceleration technique with PLD is borrowed from literature. Its technical contribution is not significant.
- The paper may need a quantitative breakdown of how often FLy’s deferred acceptance mechanism is triggered, and what fraction of accepted tokens are due to semantic mismatch tolerance rather than exact matches. Reporting the percentage of tokens where FLy overrules vanilla SPD and the resulting speedup contribution would directly substantiate the claim that loose verification is the key driver of acceleration. Without this, it remains unclear how much of the speedup is due to FLy’s core innovation versus other factors (e.g., larger K, MLA).
- The ablation studies in Table 4 vary only one hyper-parameter at a time (W, θ, K), while keeping others fixed. This one-at-a-time approach is insufficient for exploring the joint effect of hyper-parameters. A grid search over the hyper-parameter space is necessary to ensure that the reported configurations are globally optimal, not just locally tuned.

**Questions:**

- The reproducibility of the work may need improvement through open-source sharing. Would you provide your source code?
- Regarding the performance of JudgeDecoding in the paper, please detail the reproduction of the results.
- How do you explain the extremely high recovery rates (≥99.5%, including multiple 100%) reported in Figure 3, Tables 4, 6, and 7? Given that FLy explicitly accepts semantically mismatched tokens, some degradation in task-level accuracy is expected.
  - What is the standard deviation across multiple runs?
  - How many test samples were used per dataset?
  - Could you provide confidence intervals or bootstrap estimates to demonstrate that these values are statistically stable, rather than artifacts of small test sets or coarse metrics?
- Please address the issues mentioned in "Weakness".

---

> ### Author Response · Authors · 2025-11-20
> **Response to Reviewer oayV (1/4)**
>
> We thank you for your careful review and insightful feedback on our work. Unless stated otherwise, our experimental conditions in rebuttal are the same as those in the ablation study.
>
> > Weakness 1: For Figure 1, it is hard to tell whether Fly is more robust against OOD datasets as the degration rate on speedup is similar.
>
> - In our manuscript (lines 296–303), “ID/OOD datasets” are defined with respect to EAGLE-3’s training distribution. Specifically, MGSM, NIAH, and ACP in Figure 1 are all out-of-domain for EAGLE-3. Tables 2 and 3 show that EAGLE-3 achieves a high speedup of **3.83$\times$** on in-domain datasets, but its speedup drops to **1.56$\times$** on the OOD datasets. Our method FLy, being training-free, does not suffer from such a domain gap: it achieves **2.63$\times$** speedup on the in-domain datasets and **2.53$\times$** on the OOD datasets, i.e., its performance remains much more stable across the distribution shift. This consistency between ID and OOD is what we refer to as robustness.
> - Regarding Figure 1: indeed, all methods naturally experience some slowdown on more challenging datasets. However, this is largely due to intrinsic properties of the tasks, such as longer average output lengths. Such task-induced slowdowns cannot be completely removed by a training-free design alone.
>
> - Our claim is therefore not that FLy avoids any degradation on harder datasets, but that:
> FLy maintains strong and stable speedups across both in-domain and OOD datasets, while
> EAGLE-3’s performance degrades much more sharply on OOD datasets.
>
> - We clarified this perspective in the revised version in lines 361-364.
>
>
> > Weakness 2: While outperforming training-based methods on OOD data is a key result, the paper notes that FLy is slightly slower than the heavily optimized EAGLE-3 on ID datasets. An in-depth discussion or analysis of this performance gap beyond the expected trade-off for "training-free" would improve the technical quality of the paper.
>
>
> - The observed performance gap on ID data is due to the fact that EAGLE-3’s drafter is explicitly trained on the ID distribution, which yields very high draft–target alignment and thus superior efficiency.
>
> - For EAGLE-3, each speculative decoding (SPD) round has essentially fixed cost: the drafter proposes a fixed number of tokens and the target performs one forward pass to verify them. The difference between ID and OOD therefore mainly comes from the number of accepted tokens per round $\tau$. From Tables 2 and 3, with Llama-3.3-70B as the target, **EAGLE-3 accepts on average $\tau = 2.50$ tokens per round on OOD datasets, but $\tau = 5.61$ on ID datasets**. Since the per-round cost is constant, a larger $\tau$ on ID directly reduces the required SPD rounds and yields higher end-to-end speedups.
>
> - **In contrast, FLy behaves stably across distributions: it achieves $\tau = 11.35$ on OOD and $\tau = 11.87$ on ID**. To obtain such long acceptance lengths, FLy uses a separate 8B drafter and a larger draft length per round (K = 15 for 70B targets), making each SPD round more expensive than in EAGLE-3. On OOD data, **the much larger $\tau$ more than compensates for this extra per-round cost**, but on ID data, EAGLE-3’s distribution-trained drafter attains sufficiently high $\tau$ that FLy’s longer draft time becomes a net disadvantage.
>
> - To make this trade-off explicit, we report the **wall-clock time per SPD round and the derived time per accepted token**, as shown in the table below.
>   - On OOD benchmarks, FLy’s larger $\tau$ yields a lower time per token (22.64 ms vs. 34.84 ms), explaining our higher end-to-end speedups.
>   - On ID benchmarks, EAGLE-3’s ID-trained drafter reduces its time per token to 15.53 ms, now lower than FLy’s 21.65 ms, so EAGLE-3 becomes faster overall despite FLy still accepting more tokens per round. We added this analysis in Appendix J.
>
> |Method|Time per SPD round(ms)|Time for one token in OOD dataset(ms)|Time for one token in ID dataset(ms)|
> |---|---|---|---|
> |EAGLE-3|87.11|34.84|15.53|
> |FLy|256.94|22.64|21.65|

---

> ### Author Response · Authors · 2025-11-20
> **Response to Reviewer oayV (2/4)**
>
> > Weakness 3: The Multi-level acceleration technique with PLD is borrowed from literature. Its technical contribution is not significant.
>
> PLD is only one instantiation of MLA. Conceptually, MLA is a systematic way to address the new drafter-side bottleneck introduced by FLy’s high $\tau$, by reusing existing training-free accelerators.
>
>
> In FLy, by accepting semantically correct drafts, the average number of accepted tokens per round ($\tau$) becomes significantly larger than in existing SPD methods, and the 8B model is also much heavier than EAGLE-3’s one-layer transformer. As a result, **the dominant cost in each SPD round shifts from the target-model verification to the drafter’s auto-regressive generation**. This effect is explicitly quantified in Table below.
>
> |Method|Time per SPD round(ms)|Time for one token in OOD dataset|Time for one token in ID dataset|
> |---|---|---|---|
> |EAGLE-3|87.11|34.84|15.53|
> |FLy|256.94|22.64|21.65|
>
>
> The MLA mechanism is designed precisely to **address this new bottleneck** introduced by our loose verification scheme. Instead of accelerating the target model (as done in prior SPD work), **MLA accelerates the drafter itself while preserving the following properties that are central to FLy:**
> - Training-free and parameter-free design. We instantiate MLA with a PLD-style n-gram lookup, which is extremely lightweight and does not introduce any additional trainable parameters. This is crucial to **maintain FLy’s plug-and-play, training-free nature and its robustness under distribution shift**, which we emphasize as a key advantage over training-based speculative methods.
>
> - Tight integration with loose SPD. In existing PLD usage, the lookup mechanism is applied directly to the target model. In our setting, PLD is repurposed and integrated as a draft-side accelerator inside a multi-level speculative decoding pipeline. The target-level acceleration comes from FLy’s entropy gate and deferred-window mechanism, and
> the draft-level acceleration is provided by MLA to **mitigate the drafter to become the dominant latency source** when $\tau$ is large.
>
>
> We therefore view MLA as a non-trivial, system-level contribution: it is a carefully chosen, parameter-free instantiation of a more general design principle: accelerating both the target and the drafter in high-$\tau$ speculative decoding while maintaining the core goals of FLy (training-free, model-agnostic, and OOD-robust).
>
>
>
> > Weakness 4: The paper may need a quantitative breakdown of how often FLy’s deferred acceptance mechanism is triggered, and what fraction of accepted tokens are due to semantic mismatch tolerance rather than exact matches. Reporting the percentage of tokens where FLy overrules vanilla SPD and the resulting speedup contribution would directly substantiate the claim that loose verification is the key driver of acceleration. Without this, it remains unclear how much of the speedup is due to FLy’s core innovation versus other factors (e.g., larger K, MLA).
>
> (1)  We added diagnostics to quantify how often FLy’s deferred acceptance is used, under the same setup as our ablations.
>
> - We first measure **how many mismatch tokens are still accepted by FLy due to loose verification, and how many are ultimately rejected**. As shown in table below, FLy accepts about **34.3% (1,463 / 4,267)** of the tokens that vanilla SPD would reject. These decisions are precisely where FLy’s deferred acceptance mechanism overrules the strict exact-match rule.
>
> |Statistics |Number|
> |---|---|
> |mismatch tokens|4267|
> |mismatches accepted by FLy|1463|
> |mismatches finally rejected|2804|
>
> - Next, among all accepted tokens, we measure **how many are additionally accepted only because of loose verification**. These include: tokens that are themselves mismatches but accepted by FLy, and downstream matching tokens that become accepted in the same drafting round because an earlier mismatch was accepted by FLy. As shown in table below, we find that **21.6% (7,137 / 33,089)** of accepted tokens are due solely to FLy’s loose verification. This provides direct quantitative evidence that FLy frequently overrules vanilla SPD and that loose verification is heavily exercised during decoding, rather than being a rare corner-case behavior.
>
> |Statistics |Number|
> |---|---|
> |all accepted tokens|33089|
> |additional tokens accepted by FLy|7137|
>
>
>
> (2)  To **isolate the effect of loose verification from K and MLA**, we also apply MLA to vanilla SPS and fix K = 15 for both SPS and FLy. The speedup results on HumanEval are:
>
> |Method |Speedup|
> |---|---|
> |SPS(w/ MLA)|1.72$\times$|
> |FLy(w/ MLA)|2.86$\times$|
> |SPS(w/o MLA)|1.64$\times$|
> |FLy(w/o MLA)|2.69$\times$|
>
> Even without MLA and at the same K, FLy reaches **2.69$\times$ vs. 1.64$\times$** for SPS, showing that the remaining speedup gap is attributable purely to FLy’s loose verification design.
>
> We added these experiments in Appendix G and I.

---

> ### Author Response · Authors · 2025-11-20
> **Response to Reviewer oayV (3/4)**
>
> > Weakness 5: The ablation studies in Table 4 vary only one hyper-parameter at a time (W, θ, K), while keeping others fixed. This one-at-a-time approach is insufficient for exploring the joint effect of hyper-parameters. A grid search over the hyper-parameter space is necessary to ensure that the reported configurations are globally optimal, not just locally tuned.
>
>
> Our ablation study in Table 4 was originally designed to highlight the individual role and qualitative effect of each hyper-parameter. Varying one at a time allows us to clearly illustrate how each component influences the speed–accuracy trade-off, without conflating multiple factors.
>
> To address your concern more directly, we have now additionally performed a grid search over the joint hyper-parameter space, where each entry is reported as $(\text{speedup}, \tau, \text{Recovery})$:
>
> $\theta=0.3$ :
>
> |W| K=10 | K=15 | K=20 | K=25 |
> |------|------|------|------|------|
> | W=4  | ($2.84\times, 9.53, 98.2\%$) | ($2.91\times, 13.27, 97.6\%$) | ($2.91\times, 16.63, 96.9\%$) | ($2.90\times, 19.40, 95.7\%$) |
> | W=6  | ($2.75\times, 9.04, 100\%$) | **($2.86\times, 12.61, 100\%$)** | ($2.59\times, 15.60, 100\%$) | ($2.58\times, 18.56, 100\%$) |
> | W=8  | ($2.54\times, 8.50, 100\%$) | ($2.61\times, 11.96, 100\%$) | ($2.43\times, 14.99, 100\%$) | ($2.55\times, 17.23, 100\%$) |
>
> $\theta=0.6$ :
>
> | W | K=10  | K=15 | K=20 | K=25 |
> |---|---|---|---|---|
> | W=4  | ($2.78\times, 9.30, 98.7\%$) | ($2.86\times, 12.63, 98.2\%$) | ($2.75\times, 15.60, 97.6\%$) | ($2.71\times, 18.28, 96.3\%$) |
> | W=6  | ($2.68\times, 8.92, 100\%$)  | ($2.76\times, 12.20, 100\%$)  | ($2.53\times, 15.10, 100\%$)  | ($2.52\times, 17.47, 100\%$)  |
> | W=8  | ($2.50\times, 8.40, 100\%$)  | ($2.55\times, 11.68, 100\%$)  | ($2.37\times, 14.43, 100\%$)  | ($2.48\times, 16.53, 100\%$)  |
>
>
>
> We added the joint grid-search results to Appendix F in the revised version.
>
>
> > Question 1: The reproducibility of the work may need improvement through open-source sharing. Would you provide your source code?
>
> We will release the full code upon acceptance of the paper. We look forward to the community building on our work and further exploring the training-free SPD.
>
> > Question 2: Regarding the performance of JudgeDecoding in the paper, please detail the reproduction of the results.
>
> Unfortunately, as far as we are aware, JudgeDecoding is trained on a **private dataset**, and the authors **have not released their code, training data, or checkpoints**. This prevents us from reproducing their method under our experimental setup. For this reason, we do not report a direct speedup comparison with JudgeDecoding on our hardware.
>
> In Fig. 3, the JudgeDecoding results are **taken directly from the original paper** and only used for accuracy preservation comparison. We cannot directly use the reported speedup ratio because decoding speed is highly dependent on the underlying device, as also emphasized by prior work such as Spec-Bench [1]. Without an official implementation or checkpoints, any speed comparison would be unreliable and potentially misleading.
>
> In the revised version, we clarified that the JudgeDecoding results are directly copied from the original paper, as shown in line 374.
>
> [1] Spec-Bench: A Comprehensive Benchmark and Unified Evaluation Platform for Speculative Decoding.

---

> ### Author Response · Authors · 2025-11-20
> **Response to Reviewer oayV (4/4)**
>
> > Question 3: How do you explain the extremely high recovery rates (≥99.5%, including multiple 100%) reported in Figure 3, Tables 4, 6, and 7? Given that FLy explicitly accepts semantically mismatched tokens, some degradation in task-level accuracy is expected. What is the standard deviation across multiple runs? How many test samples were used per dataset? Could you provide confidence intervals or bootstrap estimates to demonstrate that these values are statistically stable, rather than artifacts of small test sets or coarse metrics?
>
> (1) Why are recovery rates so high?
> - The very high recovery rates are a direct consequence of our **intentionally conservative hyperparameter choices**, which are designed to **prioritize accuracy preservation(>99%)** over aggressive speedup:
>     - We use a strict deferred-mismatch rule with window $W=6$: on a draft–target mismatch, FLy inspects the next 6 tokens before deciding to accept or reject.
>     - We set a relatively high entropy threshold (0.3), so lossy acceptance is only allowed when the target is already uncertain.
>
>
> - This behavior is further corroborated by our ablation study: when we shorten the $W$ (e.g., from 6 to 4) or lower the entropy threshold, we indeed obtain higher speedups but the recovery rate drops to around 97%. This pattern indicates that the extremely high recovery rates in the main configuration are not artifacts of small test sets or coarse metrics, but rather the result of deliberately conservative hyperparameters that keep most accepted mismatches within a “semantically safe” region.
>
> (2) In our accuracy-preservation experiments (Figures 3 and Tables 4, 6, and 7), **we use greedy decoding with temperature=0**. Under this setting, the generation process is fully deterministic: repeated runs with the same model and data produce identical outputs. For this reason, we did not conduct multiple stochastic runs.
>
> (3) **For the 100% accuracy-preservation results**, such as those on HumanEval and NIAH:
> - Code benchmarks of this type are relatively insensitive to small perturbations, and there are often multiple distinct outputs that are all judged as correct, so maintaining 100% accuracy is unsurprising.
> - For the NIAH dataset, the draft model already achieves very high accuracy, so preserving 100% accuracy there is also expected.
>
>
> (4) To address statistical stability, we report sample counts and Wilson 95% CIs for all benchmarks, evaluated with lm-evaluation-harness [2] on the full test sets:
>
> | Dataset         | Sample number | Accuracy | Wilson 95% CI   |
> | --------------- | ------------- | ------------ | --------------- |
> | HumanEval       | 164           | 100%         | [97.7%, 100%]   |
> | GSM8K           | 1319          | 99.2%        | [98.5%, 99.5%]  |
> | MBPP            | 378           | 99.7%        | [98.5%, 100.0%] |
> | ACP prog gen    | 130           | 99.3%        | [95.8%, 99.9%]  |
> | NIAH multivalue | 500           | 100%         | [99.2%, 100.0%] |
> | MGSM          | 750           | 99.2%        | [97.1%, 99.8%]  |
>
>
> These results are now reported explicitly to Appendix H in the revised version.
>
> [2] https://github.com/EleutherAI/lm-evaluation-harness.git

---

> ### Comment · Reviewer_oayV · 2025-11-24
>
> Thanks to the authors for the careful responses and additional experiment results. I am happy with most responses except for Question 2, which still raises concern about fairness. I would like to raise the score accordingly.

---

> > ### Author Response · Authors · 2025-11-24
> >
> > Thank you very much for your thoughtful feedback and for acknowledging our efforts in addressing your comments. We sincerely appreciate your willingness to raise your score, and we’re grateful for your constructive engagement with our work.

---

### Official Review · Reviewer_RKkU · 2025-10-31

**Soundness:** 3
**Presentation:** 3
**Contribution:** 3
**Rating:** 6
**Confidence:** 4

**Summary:**

This paper studies speculative decoding with lossy verification, a topic that has recently garnered some attention in the community. The authors investigate training-free approaches that enable the target model to accept tokens it would usually reject under the standard regime, while maintaining high output quality. This is achieved through a two-stage process as follows: First, an entropy gate is designed, where the rejection of tokens that would get rejected according to the target logits, is deferred in case the entropy of the target distribution was very high, suggesting that multiple tokens could potentially be used instead of solely the most likely one. The deferred token is then put on “probation”, i.e. if another rejection within a certain window occurs within the continuation of said token, then it will be rejected. If no other rejections occur, then the token is accepted. This follows from the insight that target models tend to correct mistaken tokens in subsequent steps, making the likelihood of another rejection significantly higher in case the token truly should have been rejected.

The authors then evaluate their approach in the setup of draft / target speculative decoding, with models of size roughly 8B serving as the draft for targets of size 70B and 405B. The proposed approach is evaluated on a range of benchmarks and compared against recent models such as Eagle-3. On out-of-distribution datasets, the novel approach outperforms the baselines and remains competitive in-distribution as well, while maintaining the output quality.

**Strengths:**

1. Lossy verification in speculative decoding is a timely topic that enables more efficient decoding for large models. Making progress in this area is very relevant for the field to further push the efficiency gains.
2. The paper is very nicely written and easy to understand. The proposed approach is very well motivated and the experiments are thorough, which is needed to assess its validity.
3. The technique doesn’t require any training and any model can easily be enhanced with this approach, making it easy for users to experiment with it for their own model.

**Weaknesses:**

1. The out-of-distribution numbers are quite strong but in-distribution, training-based approaches like EAGLE-3 still keep the upper hand, with quite a discrepancy. What seems a bit problematic however in this experiment design is that different speculative decoding techniques are compared; standard small / large model speculation (Fly) and self-speculation (EAGLE-3) where the draft model is a very small model on top of the features. Usually, more involved like tree-attention are also involved. Thus enhancing small / large model speculation with the new techniques might not be the best comparison. Why not also equip EAGLE-3 with the same techniques as introduced in this paper, or am I missing an incompatibility? Indeed, the approach presented here does outperform the standard small / large model speculation setup without these enhancements.
2. I could not find any tokens/s metrics, arguably the most important metric in LLM inference. What frameworks are the experiments based on in the main paper? I found some results in vLLM in the appendix but they’re unfortunately very limited. I find experiments in this setting way more important as they actually reflect real scenarios more accurately. Many conclusions in the literature have changed in the past when moving to more optimized frameworks, so it would be great to verify that the proposed approach remains superior even in more optimized settings like in vLLM.

**Questions:**

1. When looking for later mistakes in the window, you seem to not apply the entropy rule anymore, i.e. any direct mismatch counts as a mistake within the window, correct?
2. The section on Multi-Level Acceleration confused me a bit initially. Of course when drafting longer candidate sequences with high acceptance rates, the generation process starts to spend most time on drafting. But from a pure speculative decoding perspective, this is of course inevitable (in the extreme case we draft the entire sequence once and accept everything with the target in a single forward pass). I would state more clearly that MLA is an idea to further speed-up the entire process, but it’s not a mechanism to improve the speculative decoding setup you’re studying, it’s a way to speed-up drafting itself. Or did I misunderstand something?
3. Can one equip EAGLE-3 with your augmentations to enable it to have lossy verification? In general, does the approach extend to any speculative decoding setup?

---

> ### Author Response · Authors · 2025-11-20
> **Response to Reviewer RKkU(1/3)**
>
> We sincerely appreciate your thoughtful comments and suggestions. We have provided detailed responses to your questions below to address your concerns. Unless stated otherwise, our experimental conditions in rebuttal are the same as those in the ablation study.
>
> > Weakness 1: The out-of-distribution numbers are quite strong but in-distribution, training-based approaches like EAGLE-3 still keep the upper hand, with quite a discrepancy. What seems a bit problematic however in this experiment design is that different speculative decoding techniques are compared; standard small / large model speculation (Fly) and self-speculation (EAGLE-3) where the draft model is a very small model on top of the features. Usually, more involved like tree-attention are also involved. Thus enhancing small / large model speculation with the new techniques might not be the best comparison. Why not also equip EAGLE-3 with the same techniques as introduced in this paper, or am I missing an incompatibility? Indeed, the approach presented here does outperform the standard small / large model speculation setup without these enhancements.
>
> (1) In-domain performance: Our method is fully training-free, whereas EAGLE-3 is a training-based approach optimized directly on its in-domain benchmarks. On such ID datasets, it is therefore expected that our training-free FLy still lags behind EAGLE-3 in speedup. Narrowing this gap while preserving FLy’s robustness on OOD data is an important direction for future work.
>
> (2) FLy is orthogonal to EAGLE-3 and can in principle be combined with it. We report experiments on “EAGLE-3 + FLy” in the answer to Question 3. However, this setting does not fully highlight FLy’s strengths:
>
>
> - FLy explicitly exploits the target LLM’s self-correction over a deferred window with length $W$, which requires draft sequences substantially longer than $W$. We therefore pair FLy with an 8B drafter that can reliably generate 15–25 coherent tokens. By contrast, EAGLE-3 uses a very shallow self-speculative drafter (one Transformer layer) with draft depth 6. When we increased its draft length to 15–25 tokens, draft quality degraded and the target rejected many more tokens, so simply stacking FLy on top of the original EAGLE-3 setup is not ideal.
> - A core motivation of our work is to improve the robustness of SPD in OOD settings. EAGLE-3 is a training-based method and it exhibits clear degradation on OOD benchmarks. Even if we equip EAGLE-3 with FLy, its behavior on OOD data would still be fundamentally limited by this domain-gap issue, which FLy alone cannot eliminate. Meanwhile, the standard small/large speculation setup we adopt is completely training-free and therefore does not suffer from training-induced distribution shift.
>
>
> We clarified these points in the revised version in lines 448-454.

---

> ### Author Response · Authors · 2025-11-20
> **Response to Reviewer RKkU (2/3)**
>
> > Weakness 2: I could not find any tokens/s metrics, arguably the most important metric in LLM inference. What frameworks are the experiments based on in the main paper? I found some results in vLLM in the appendix but they’re unfortunately very limited. I find experiments in this setting way more important as they actually reflect real scenarios more accurately. Many conclusions in the literature have changed in the past when moving to more optimized frameworks, so it would be great to verify that the proposed approach remains superior even in more optimized settings like in vLLM.
>
> **(1) vLLM experiments.** we conducted experiments on LLaMA-3.3-70B-Instruct with vLLM, and report detailed tokens/s metrics for our method and the baselines:
>
> |Method|Batchsize=2|BS=4|BS=8|BS=16|
> |---|---|---|---|---|
> |Auto-regressive|45.02 toks/s(1$\times$)|82.41 toks/s(1$\times$)|105.2 toks/s(1$\times$)|194.3 toks/s(1$\times$)|
> |FLy|103.55 toks/s(2.30$\times$)|163.17 toks/s(1.98$\times$)|194.62 toks/s(1.85$\times$)|332.25 toks/s(1.71$\times$)|
> |EAGLE-3|82.79 toks/s(1.84$\times$)|167.38 toks/s(2.03$\times$)|287.9 toks/s(2.74$\times$)|474.63 toks/s(2.44$\times$)|
>
> In vLLM, our speedup is higher than EAGLE-3 at small batch sizes but then decreases as the batch size grows. This is because, under vLLM, the inference of our 8B draft model is highly optimized, which greatly reduces the time difference between our drafter and that of EAGLE-3. As a result, when the batch size is small, our method can achieve higher end-to-end throughput than EAGLE-3. However, as the batch size increases, the 8B draft model has to process much larger batches, which causes its speed to degrade more noticeably. In contrast, the draft model used by EAGLE-3 is much smaller, so its wall-time is less sensitive to the batch size and varies less across different batch-size settings.
>
> These results are included in the Appendix D in the revised version.
>
> **(2) Frameworks used in the main paper.** The main experiments are implemented using the HuggingFace Transformers library. The current vLLM implementation supports only a limited subset of algorithms, several training-based baselines (such as EAGLE-2), some tricks(such as tree-attention methods) and most training-free methods(such as REST and TokenRecycling) are not supported in vLLM. To enable a fair and comprehensive comparison, we therefore report the main results using Transformers implementations and use vLLM as a complementary system-level evaluation.
>
>
> > Question 1: When looking for later mistakes in the window, you seem to not apply the entropy rule anymore, i.e. any direct mismatch counts as a mistake within the window, correct?
>
> Yes, you are correct. When we encounter a deferred mismatch, we then examine the subsequent window_length ($W$) tokens. If any of these tokens exhibits a direct mismatch, we immediately reject the deferred mismatch. In this check within the window, we do not apply the entropy rule again, the decision is purely based on whether a mismatch occurs.
>
> > Question 2: The section on Multi-Level Acceleration confused me a bit initially. Of course when drafting longer candidate sequences with high acceptance rates, the generation process starts to spend most time on drafting. But from a pure speculative decoding perspective, this is of course inevitable (in the extreme case we draft the entire sequence once and accept everything with the target in a single forward pass). I would state more clearly that MLA is an idea to further speed-up the entire process, but it’s not a mechanism to improve the speculative decoding setup you’re studying, it’s a way to speed-up drafting itself. Or did I misunderstand something?
>
>
> - MLA is indeed orthogonal to the verification scheme and is specifically designed to speed up the drafting phase itself rather than to change the verification protocol.
> - It is particularly beneficial in our setting, where the algorithm can safely propose long draft sequences, making drafting time rather than target-model verification time the primary bottleneck (80.6% of the total time for FLy, vs. 21.3% for EAGLE-3; see Table below). This stands in clear contrast to prior approaches such as EAGLE-3, where the target model's verification phase often bottlenecks the runtime.
>
>
> |Method|Draft time(ms)|Total time (ms) |Draft time / total time|
> |---|---| ---| ---|
> |EAGLE-3| 18.52|87.11|21.26% |
> |FLy(ours)| 245.51|304.58|80.6% |
>
> - This behavior highlights an inherent upper bound: even in the ideal case where all drafted tokens are accepted, one must at least pay the cost of generating these tokens with the drafter. Our MLA mechanism is specifically aimed at reducing this drafting cost. By using a training-free PLD-style prediction scheme to further accelerate the drafter, we shorten the drafting phase and effectively alleviate this bottleneck for our method.
>
>
>
> We clarified this perspective in the revised version in lines 251-255.

---

> ### Author Response · Authors · 2025-11-20
> **Response to Reviewer RKkU (3/3)**
>
> > Question 3: Can one equip EAGLE-3 with your augmentations to enable it to have lossy verification? In general, does the approach extend to any speculative decoding setup?
>
> - **In principle**, our lossy verification scheme can indeed be combined with EAGLE-3, and more generally with any speculative decoding method (as long as the method exposes (i) the drafted token sequence and (ii) the target model’s logits for verification), because our approach only modifies the verification rule on the target side and is orthogonal to how draft tokens are generated. We have implemented **EAGLE-3 + Loosely Verify** and included the corresponding results in the experiments as shown in the table below and also in Table 9 in the revised paper.
>
> - **In practice**, however, there are practical constraints that affect how much benefit one can obtain.
>   - Our method explicitly exploits the LLM’s self-correction ability by deferring a mismatch and then examining the following W=6 tokens. We decide to reject the current mismatch based on the appearance of any error in this future window. **This naturally requires that each draft step produces more than W tokens**; otherwise, there is very limited room to apply lossy verification.
>   - In our setting, we use an 8B draft model, which can reliably generate longer draft segments (e.g., **15 or 25 tokens**) that remain coherent and logically consistent. However, many self-speculative methods, including EAGLE-style approaches, typically use a relatively small draft depth (e.g., **4-6 tokens**). To fully leverage our scheme, one would need to draft substantially more tokens per step, which is challenging for their very lightweight drafter. when we increase its draft length to 15–25 tokens, the draft quality deteriorates, and the drafted tokens are rejected much more frequently by the target model.
>
>
>
>
> - **Experiments on EAGLE-3 + FLy:**
> Since FLy needs to inspect the W = 6 tokens following a mismatch, while the original EAGLE-3 configuration uses a draft depth of exactly 6, EAGLE-3 in its default setting cannot directly adopt FLy. We therefore extend EAGLE-3’s draft depth so that it composes better with our method.
>
>
>
> |Setting|Draft depth|speed up|$\tau$|Recovery|
> |---|---|---|---|---|
> |Original EAGLE-3 (W/o FLy)|6|3.97 $\times$|5.73|100%|
> |EAGLE-3 (W/ FLy)|10|4.17$\times$|6.42|100%|
> |EAGLE-3 (W/ FLy)|15|3.99$\times$|6.48|100%|
>
> We added the experiments in the revised version in lines 448-454.

---

### Official Review · Reviewer_6AHf · 2025-11-01

**Soundness:** 3
**Presentation:** 3
**Contribution:** 3
**Rating:** 8
**Confidence:** 2

**Summary:**

The authors propose a novel method to accelerate LLMs via speculative decoding. In particular, they relax the rigid assumption of verification in classic speculative decoding and propose a two-tier mechanism. The method comprises first an entropy gate to evaluate whether a mismatch should be directly rejected, and then a token-level defer window to find whether the mismatch is benign. The efficacy of the method is showcased on multiple benchmarks and across different draft and target models.

**Strengths:**

- The main idea is novel and clearly presented.

- The studied problem is relevant and the proposed solution appropriately motivated.

- There is a plethora of experimental evidence across different models that support the efficacy of the proposed method.

- The proposed method is benchmarked against mutliple SOTA baselines and showcases superior performance while requiring no training.

- The comparison with learning-based approaches, i.e., judge-decoding, is insightful and highlights the capabilities of the proposed method.

- I appreciate the thorough ablations on cross-model drafting.

**Weaknesses:**

Although I do not see any major flaws, there are some issues that I believe would worth addressing:

- I am missing some discussion and/or analysis of the shortcomings and failure modes of the proposed model.

- For completeness, it would be interesting to see what are the speedups achieved by the proposed and judge-decoding in Fig. 3.

- I am missing some related work, e.g., [1, 2]

 [1]. Learning harmonized representations for speculative sampling

 [2]. Glide with a cape: A low-hassle method to accelerate speculative decoding

**Questions:**

I would appreciate if the authors address the questions raised in the weaknesses section.

---

> ### Author Response · Authors · 2025-11-20
> **Response to Reviewer 6AHf**
>
> We sincerely appreciate your thoughtful comments and suggestions. We have provided detailed responses to your questions below to address your concerns.
>
> > Weakness1: I am missing some discussion and/or analysis of the shortcomings and failure modes of the proposed model.
>
> - Shortcomings: Our approach is fully training-free, whereas methods such as EAGLE-3 are training-based and are optimized on their in-domain benchmarks. As a consequence,  on benchmarks that closely match these training domains, our method may underperform EAGLE-3. Bridging this gap while preserving the training-free nature of our method is an interesting direction for future work.
> - Failure modes: Our method is designed to preserve semantic consistency with the target model, rather than enforcing exact token-level agreement. This design works well for most natural-language generation scenarios, but it can be suboptimal on tasks that require exact reproduction at the token level. Examples include faithfully repeating a long passage verbatim or reciting a poem where every character must match the reference. For such tasks, the loose verification mechanism is inherently less suitable, since the evaluation requires exact token-level reproduction.
> In such cases, we recommend setting the entropy thresholds to 1 (see Table 4), transforming FLy to a typical SPS configuration and trading semantic flexibility for exact token matching.
>
> We add a Limitations section in the revised version, where we explicitly discuss these shortcomings and failure modes.
>
> > Weakness 2: For completeness, it would be interesting to see what are the speedups achieved by the proposed and judge-decoding in Fig. 3.
>
> - Unfortunately, as far as we are aware, JudgeDecoding is trained on a **private dataset**, and the authors **have not released their code, training data, or checkpoints**. This prevents us from reproducing their method under our experimental setup. For this reason, we do not report a direct speedup comparison with JudgeDecoding on our hardware.
>
> - In Fig. 3, the JudgeDecoding results are **taken directly from the original paper** and only used for accuracy preservation comparison. We cannot directly use the reported speedup ratio because decoding speed is highly dependent on the underlying device, as also emphasized by prior work such as Spec-Bench[3]. Without an official implementation or checkpoints, any speed comparison would be unreliable and potentially misleading.
>
> In the revised version, we clarified that the JudgeDecoding results are directly copied from the original paper as shown in line 374.
>
> [3] Spec-Bench: A Comprehensive Benchmark and Unified Evaluation Platform for Speculative Decoding.
>
>
>
>
> > Weakness3: I am missing some related work, e.g., [1, 2]
>
>
> - HASS (Zhang et al., 2024) introduces harmonized training objectives and context alignment to reduce discrepancies between training and inference in speculative decoding, leading to improved acceptance rates without additional inference-time cost.
>
> - GLIDE (Du et al., 2024) accelerates speculative decoding by allowing the draft model to reuse the target LLM’s KV cache via cross-attention, and by dynamically expanding proposals based on token-level confidence.
>
> Both are training-based speculative decoding methods that require retraining, whereas FLy is fully training-free and can be plugged into existing LLMs without additional training. We have added a discussion of these works to the Related Work section in the revised manuscript (Lines 479–481)

---

### Author Response · Authors · 2025-12-01
**To AC: Summary of the rebuttal and revisions**

Dear AC,

We sincerely thank all reviewers for their insightful and constructive comments. We have carefully revised the manuscript and addressed all concerns raised during the review process. We fully recognize the significant responsibilities and workload of the AC. To assist with your decision-making, we have prepared a concise summary of our rebuttal and the revisions made.

The four reviewers raised a total of 22 weaknesses and questions, covering the following areas:
1. **Experiments (9)**:
    - Ablation Studies (3):
        - A rigorous grid search to isolate the speedup contributions of loose verification versus Multi-Level Acceleration (Reviewer oayV).
        - Additional ablation studies on the proposed MLA (Reviewers oayV, Eqon).

    - Analysis Experiments (3):
        - Real-world tokens/s metrics on vLLM (Reviewer RKkU).
        - Statistical confidence intervals to validate the high recovery rates (Reviewer oayV).
        - A quantitative breakdown of FLy's loose verification mechanism (Reviewer oayV).

    - Integration with Existing Methods (3):
        - Demonstration of FLy's compatibility with Tree-Verification (Reviewer Eqon).
        - Evaluation of FLy in combination with EAGLE-3 (Reviewer RKkU).

2. **Further Explanation (5)**:
    - Clarification of robustness under OOD settings (Reviewer oayV).
    - Expanded explanation of the motivation behind MLA (Reviewers oayV, RKkU).
    - Additional insights into the loose verification mechanism (Reviewer RKkU).

3. **Additional Discussion (8)**:
    - Analysis of failure modes and task sensitivity (Reviewers 6AHf, Eqon).
    - Corrections to the related work section (Reviewers 6AHf, Eqon).
    - Clarifications regarding the comparison with JudgeDecoding (Reviewers 6AHf, oayV).
    - Confirmation of the plan to release the code (Reviewers oayV, Eqon).


We have added comprehensive new experiments and clarifications. All experimental results match the expectations, fully addressing all concerns. Notably, Reviewer oayV explicitly acknowledged our response and **raised their score from 6 to 8**. The other reviewers provided no further comments.

Regarding the strengths of our work, reviewers recognized the novelty and efficacy of the proposed method, specifically praising the "two-tier mechanism" as intuitive and empirically robust (Reviewers Eqon, oayV, 6AHf). They highlighted the paper's clear motivation (Reviewers 6AHf, RKkU) and superior performance (Reviewers RKkU, oayV). Additionally, all reviewers commended the method's model-agnostic, plug-and-play nature, as well as the clarity of the presentation.


To ensure that the AC and all reviewers can easily verify the revisions, all changes have been thoroughly documented in the revised version and are clearly marked in blue.

Thank you once again for your time and consideration.

Warm regards,

Authors of Submission 1830

---

### Meta-Review · Area_Chair_C9yV · 2026-01-06

**Summary:**

This paper proposes Training-Free Loosely Speculative Decoding, a novel speculative decoding framework that relaxes strict token-level verification by introducing an entropy-based gate and a deferred verification window. Across reviewers, there is strong consensus that the problem is timely and important, the core idea is novel, and the method is clearly presented and well motivated. The approach is training-free, model-agnostic, and easy to integrate, which reviewers repeatedly highlight as a key practical advantage over training-based speculative decoding methods.

There are some concerns on (1) comparisons with training-based or self-speculative methods, (2) the role and novelty of the multi-level acceleration (MLA/PLD) component, and (3) the lack of initial reporting of system-level metrics and diagnostic statistics. Reviewers also raised concerns about related-work coverage and task sensitivity to relaxed verification. These issues were largely addressed in the rebuttal: the authors clarified the orthogonality between FLy and EAGLE-style methods, added combined experiments where feasible, disentangled the impact of loose verification from MLA, corrected related-work inaccuracies, and provided detailed statistics on mismatch acceptance, recovery rates, and confidence intervals. While MLA itself is borrowed from prior work, the reviewers generally accept it as a reasonable system-level complement rather than the core contribution.

After the rebuttal, I feel the paper presents a clear, well-supported contribution that advances the understanding and practice of speculative decoding, particularly in training-free and OOD-robust settings. Remaining concerns are relatively minor and mostly about scope, positioning, or further extensions rather than correctness. The consensus leans positive, with multiple reviewers explicitly indicating acceptance or willingness to raise scores after the response.

**Reviewer Concerns:**

see the summary above.

**Reviewer Scores:**

The reviewers already reached a consensus of acceptance.

---

### Decision · Program_Chairs · 2026-01-26

Accept (Poster)